# Mechanism of pore opening in the calcium-activated chloride channel TMEM16A

Andy K. M. Lam [ID] [1✉] & Raimund Dutzler [ID] [1✉]

The anion channel TMEM16A is activated by intracellular $Ca^{2+}$ in a highly cooperative process. By combining electrophysiology and autocorrelation analysis, we investigated the mechanism of channel activation and the concurrent rearrangement of the gate in the narrow part of the pore. Features in the fluctuation characteristics of steady-state current indicate the sampling of intermediate conformations that are successively occupied during gating. The initial step is related to conformational changes induced by $Ca^{2+}$ binding, which is ensued by rearrangements that open the pore. Mutations in the gate shift the equilibrium of transitions in a manner consistent with a progressive destabilization of this region during pore opening. We come up with a mechanism of channel activation where the binding of $Ca^{2+}$ induces conformational changes in the protein that, in a sequential manner, propagate from the binding site and couple to the gate in the narrow pore to allow ion permeation.

[1] Department of Biochemistry, University of Zurich, Winterthurerstrasse 190, CH-8057 Zurich, Switzerland. ✉email: a.lam@bioc.uzh.ch; dutzler@bioc.uzh.ch

Ligand-dependent gating is a tightly regulated process in ion channels whose open probability increases in response to the binding of an agonist to a specific site of the protein. Depending on the class of channels, the chemical nature of such ligands may range from ions to small molecules or even proteins. Ligand binding generally triggers a conformational change in the protein around the binding site, which is propagated to the pore region to open a gate that impedes ion conduction in the closed state of the channel. Gating is frequently found to be a cooperative process that proceeds via a defined set of intermediate states from the closed to the open conformation of the protein. Whereas these intermediates are transient and thus often escape structural characterization, they were in many cases successfully characterized by kinetic analysis of single-channel recordings[1–3].

TMEM16A is a ligand-dependent anion-selective channel that is activated by an increase in the intracellular $Ca^{2+}$ concentration[4–6]. The protein harbors two pores that are each contained within a single subunit of the homodimeric protein[7–9]. Both pores act independently and are activated by the binding of two $Ca^{2+}$ ions to a conserved site that is located within each subunit in proximity to the ion permeation path[10,11]. The location of the site within the membrane confers voltage dependence to the binding step and the proximity of bound $Ca^{2+}$ to the pore shapes the electrostatic potential for conduction[12]. Besides the change of pore electrostatics, the binding of $Ca^{2+}$ also triggers a conformational change in a membrane-spanning helix of the protein (α6) that coordinates the bound divalent cations after its rearrangement[8]. The movement of α6 is in turn coupled to the release of a steric gate in the narrow neck of an hourglass-shaped pore. An accompanying study[13] has identified this gate to be composed of three hydrophobic residues that are located at the intracellular end of the narrow neck. Whereas major factors controlling ion flow have been identified in previous studies and a general model for activation was proposed[14], the detailed sequence of events and the existence of intermediates in the gating process, which together define the activation mechanism, are still elusive. Although this process would ideally be characterized by single-channel analysis, such studies are prohibited by the low conductance of TMEM16A[10,15].

Here, we combine macroscopic electrophysiological measurements and autocorrelation analysis to investigate the mechanism of channel activation. We show that the fluctuation characteristics of steady-state current in TMEM16A is consistent with the sampling of intermediate conformations that are successively occupied during gating. Our results suggest a mechanism of channel activation where the binding of $Ca^{2+}$ induces conformational changes in the protein that, in a sequential manner, propagate from the binding site and couple to the gate in the narrow pore to allow ion permeation.

## Results

**Autocorrelation analysis.** Gating in ion channels is defined as the transition between non-conducting and conducting states, which is usually associated with conformational changes of the protein. Whereas gating events can be readily observed in single-channel recordings for channels of large conductance, they are hidden in the noise for those with low conduction rate such as TMEM16A[10,15]. Nonetheless, stochastic transitions in an ensemble of independent single channels give rise to fluctuations around the mean of the macroscopic steady-state current[16,17] (Supplementary Fig. 1a). The statistical properties of these fluctuations may be quantified from its power spectrum[18–21], which is the Fourier transform of the autocorrelation function of the current[22]. For single-channel fluctuations, the autocorrelation function is characterized by an exponential decay with time

constants corresponding to the system's relaxation times[19,20]. Time intervals shorter than the relaxation times are expected to yield higher correlation, as it is more likely that the channel remains in or resamples the open state, while the correlation becomes zero at much longer times as any co-occurrence of opening events separated by these time intervals is essentially random.

We illustrate these properties by calculating the autocorrelation function and power spectrum from simulated single-channel trajectories using various gating models (Supplementary Fig. 1). In all cases, the autocorrelation function consists of a sum of exponentials and the power spectrum, being the Fourier transform of the former, is a sum of Lorentzian components (Supplementary Fig. 1b, c). The number of such components defines the minimum number of transitions between states in the underlying mechanism and their respective amplitudes and corner frequencies are determined by the corresponding rate constants. Because the power spectrum of a macroscopic current is a multiple of the spectrum of a single-channel record for independent and identical channels, analysis of the former is equivalent to analyzing its single-channel counterpart in Fourier space, from which model parameters may be estimated (Supplementary Figs. 2 and 3 and "Methods"). The effect of non-stationarity (Supplementary Fig. 4) and an examination of the parameter estimation process are discussed in the methods. In this study, we applied this analysis to macroscopic currents of TMEM16A to investigate ligand binding and the subsequent transition into transient and stable conformational states. We also explored the relationship of the observed states to structural determinants during channel activation.

**Mechanism of $Ca^{2+}$ activation.** Recent cryo-EM structures of TMEM16A have revealed a conformational change involving the rearrangement of a pore-lining helix (α6) upon $Ca^{2+}$ binding in a process that precedes pore opening[8]. To gain mechanistic insight into this process and into the sequence of events following $Ca^{2+}$ binding, we used the described kinetic analysis to identify additional conformational transitions that escaped structural characterization. For that purpose, we obtained a family of power spectra by recording steady-state currents over a range of $Ca^{2+}$ concentrations, as shown in Fig. 1a, b. Evident from the trajectories in the time domain is that slow fluctuations (manifested in Lorentzian components with lower corner frequencies) vanish and fast fluctuations (those with higher corner frequencies) become more prominent as the $Ca^{2+}$ concentration increases (Fig. 1a, b). The ligand dependence of the slow fluctuations and their inverse correlation with saturation hint at events related to ligand association and dissociation. A shift in the spectral frequencies as $Ca^{2+}$ concentration is elevated is consistent with the hastening of the activation response time in concentration[8,23,24] and voltage jump experiments[25], which are governed by the same set of time constants. At saturating $Ca^{2+}$ concentrations, the spectrum is reduced to reflect fluctuations that are caused solely by gating transitions as $Ca^{2+}$ remains bound with very high probability (Fig. 1a–c). The presence of three Lorentzian components under saturating conditions (as judged by Akaike's information criterion (AIC) of models with a different number of components[26], Supplementary Fig. 5) indicates the sampling of at least four conformational states when the channel is fully occupied by $Ca^{2+}$. In conjunction with known structures, where the $Ca^{2+}$-bound conformation might resemble a ligand-activated pre-open state, our data thus point towards the presence of intermediate gating steps that relay α6 activation to the opening of the pore.

The above results motivated us to construct a kinetic model where activation of a single TMEM16A pore proceeds in three steps, traversing different conformational intermediates. In these

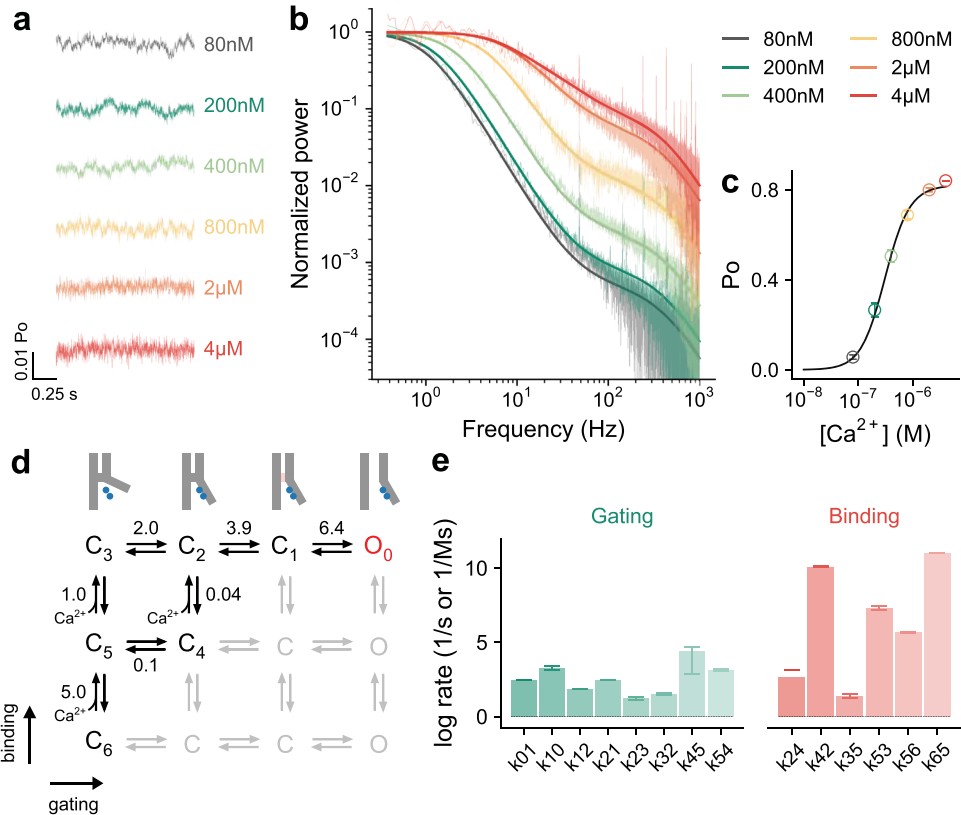

**Fig. 1 Mechanistic analysis of power spectra for wild-type TMEM16A. a** Representative section of the steady-state currents recorded at the indicated intracellular $Ca^{2+}$ concentrations at +80 mV. The data correspond to $\overline{P_O(t)} = \overline{P_O} \frac{I(t)}{I} = \overline{P_O} \frac{NiP_o(t)}{Ni\overline{P_o}}$, where $I$ is the macroscopic current, $P_o$ the open probability, $N$ the number of channels, $i$ the unitary current, and the bar notation indicates the mean of. For display, the current traces were filtered at 2.5 kHz using a digital 4-pole Bessel lowpass filter. **b** Power spectra of the respective steady-state currents shown in **a**. Data are averages of the indicated number of patches (80 nM, $n = 8$; 200 nM, $n = 8$; 400 nM, $n = 8$; 800 nM, $n = 8$; 2 μM, $n = 8$; 4 μM, $n = 7$). **c** Concentration-response relation. Data are averages of eight patches, errors are SEM. **b**, **c** Solid lines are the best-fit from global optimization using the model shown in **d** and Supplementary Fig. 2 described by Eqs. 12 and 16. **d** Minimal gating model used to analyze the power spectra and concentration dependence. Best-fit values of forward equilibrium constants and equilibrium dissociation constants (μM) are shown. **e** Kinetic parameters estimated from model fitting. Bars indicate the best-fit values of the averaged data shown in **b** and **c**. Errors are 95% confidence intervals.

states α6 is either mobile and disengaged as found in the $Ca^{2+}$-free structure of the channel ('α6-loose'), or rigid and in contact with the binding site as displayed in the $Ca^{2+}$-bound structure ('α6-tight'). From the 'α6-tight' conformation, the protein transits into another partly activated but still non-conductive state ('pre-open') before reaching the conductive open state (Supplementary Fig. 2). Three different $Ca^{2+}$ occupancies (0, 1, and 2 $Ca^{2+}$) give rise to 12 hypothetical states including three discrete open states reflecting the influence of bound $Ca^{2+}$ on conduction, which was previously observed for mutants showing pronounced basal activity[8,12]. All states are related by an allosteric Monod-Wyman-Changeux (MWC) mechanism[27,28]. In light of the low basal activity of wild type (WT) and the high cooperativity of activation, a reduced version of the model where only seven states are explicitly included was used in our subsequent kinetic analysis (Fig. 1d and Supplementary Fig. 2). Of the resulting 14 kinetic parameters, 10 ($k_{01}$, $k_{12}$, $k_{21}$, $k_{23}$, $k_{32}$, $k_{42}$, $k_{35}$, $k_{53}$, $k_{54}$, $k_{56}$) were determined independently by fitting the data (Fig. 1e). The rate $k_{65}$ describing the binding step of the first $Ca^{2+}$ was assumed to be diffusion-limited. The values of $k_{24}$ and $k_{45}$ are defined by microscopic reversibility and the empirically estimated limiting $Ca^{2+}$ binding affinity obtained from an accompanying manuscript[13]. $k_{10}$, the final opening rate constant was determined using knowledge of the maximum open probability of the channel ($P_o^{max}$) obtained from non-stationary noise analysis (Supplementary Fig. 6). The resulting equilibrium constants relating to the connected states

($L_{10}$, $L_{21}$, $L_{32}$, $L_{42}$, $L_{53}$, $L_{54}$, $L_{65}$) were obtained from the ratio of corresponding rates.

By simultaneously considering the concentration-response relation, we performed a global fit of the entire set of power spectra to the described model where $Ca^{2+}$ binding transitions are explicitly included (Fig. 1b, c). The wealth of data obtained from six different $Ca^{2+}$ concentrations ensured a unique fit of the kinetic parameters (Fig. 1e and Supplementary Table 1). Consistent with classical ligand activation models, the mechanism involves a cycle that couples affinity increment and transition efficacy (Fig. 1d) and can be viewed as the mechanistic counterpart of the conformational transition between apo and $Ca^{2+}$-bound TMEM16A observed in the structures, which is also governed by ligand binding. The affinity of binding of the second $Ca^{2+}$ ($L_{53}$) increases from ~1 μM in the resting state to ~40 nM ($L_{42}$) in the next closed state (Fig. 1d), consistent with the origin of this conformational transition being α6 activation. The rate of binding of the second $Ca^{2+}$ to the singly bound resting state ($C_5$, rate constant $k_{53}$) is approximately three orders of magnitude slower than the binding of the first $Ca^{2+}$ to the apo state ($C_6$, described by the diffusion-limited rate constant $k_{65}$), but becomes two orders of magnitude faster in the next closed state ($C_4$, $k_{42}$) (Fig. 1e and Supplementary Table 1), which, again, is consistent with the described mechanism of α6 activation where additional hydrophilic and negatively charged residues on α6 get into direct contact with the bound $Ca^{2+}$ (ref. [8]). Such electrostatically

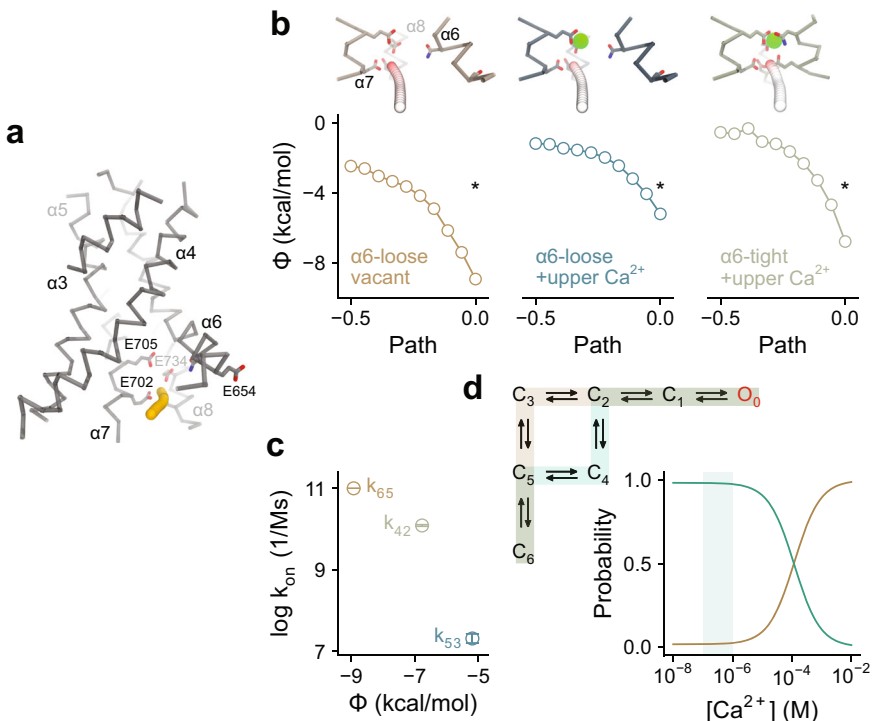

**Fig. 2 Electrostatic profiles of the Ca²⁺ binding site and their effect on the activation sequence. a** Pore region of a TMEM16A subunit in the Ca²⁺-free conformation (PDB: 5OYG) in Cα representation. Ca²⁺-binding residues are shown as sticks. Yellow spheres indicate the points at which the electrostatic potential is plotted. **b** Electrostatic potential (Φ) along the path for the indicated channel configurations (insets). Left, α6-loose conformation in the absence of Ca²⁺. Center, α6-loose conformation with the upper Ca²⁺ binding site occupied. Right, α6-tight conformation with the upper Ca²⁺ binding site occupied. Green spheres correspond to bound Ca²⁺ ions and small spheres display the trajectory at which the electrostatic potential was plotted (colors indicate the local electrostatic potential). The data point at zero (*) on the x-axis corresponds to the position at the vacant lower Ca²⁺ binding site. **c** Relationship between the calculated electrostatic potential and the estimated $k_{on}$ for Ca²⁺ binding (displayed in Fig. 1e). Errors are 95% confidence intervals. **d** Relative probability of the two possible activation sequences (top, highlighted in brown and green) as a function of Ca²⁺ concentration (bottom), calculated using Eqs. 28 and 29 with the parameters estimated from Fig. 1. The shaded region corresponds to physiologically relevant intracellular Ca²⁺ concentrations.

assisted association is also evident in Poisson-Boltzmann calculations, where the electrostatic potential of the lower binding site becomes more negative when α6 assumes an activated conformation (Fig. 2a, b). The close relationship between the electrostatic potential at the binding site and the estimated association rate constants of the respective configurations (Fig. 2c) emphasizes the correspondence of the proposed mechanism to structural states. A thermodynamic consequence of this process is the 34-fold increase in the efficacy of α6 activation when two Ca²⁺ are bound ($L_{32}$) compared to that of the mono-liganded state ($L_{54}$) (Fig. 1d and Supplementary Table 1), thus explaining the high cooperativity of TMEM16A activation.

Based on the estimated parameters, we examined the most probable sequence of conformational changes upon Ca²⁺ binding (Fig. 2d). As there is no detectable basal activity and the incorporation of a single apo closed state is sufficient to account for the data, the binding of the first Ca²⁺ most likely occurs in the state where α6 is in its resting conformation ($C_6$). The route taken by the channel is then dependent on Ca²⁺ concentration due to the Ca²⁺ dependence of the $L_{53}$ transition. In the physiological range (at sub-micromolar concentrations), the transition of α6 into a tight conformation occurs mainly through the singly bound state ($C_5$ to $C_4$) until the Ca²⁺ concentration is high enough to allow the second Ca²⁺ binding step to 'compete' for the singly bound state ($C_5$ to $C_3$) (Fig. 2d). Once two Ca²⁺ ions are bound and α6 is in its activated conformation ($C_2$), the channel can then undergo the 'gating steps', which involve a pre-open state ($C_1$) and a final transition that opens the pore ($L_{10}$).

**Rearrangement of the gate region during activation.** Equipped with the described framework for the kinetic analysis of macroscopic recordings, we sought to understand the mechanistic origin of the gating steps (i.e., the steps connecting $C_3$, $C_2$, $C_1$, and $O_0$). To this end, we combined spectral and double-mutant cycle analysis[29–32] involving residues at the inner gate (consisting of Ile 550, Ile 551, and Ile 641) (Fig. 3, Supplementary Fig. 7 and Supplementary Table 2) that give rise to constitutive activity when mutated to alanine as described in an accompanying paper[13]. In non-stationary noise analysis, we observed that only the alanine mutation of Ile 641, which forms the core of the gate, gives rise to a substantial increase in $P_o^{max}$ in the Ca²⁺-bound state, while I550A displays a modest increase and I551A a moderate decrease in $P_o^{max}$ (Fig. 3a, c). In the case of I641A and I550A, this provides additional evidence for the stabilization of the open pore conformation by the respective mutations. Analysis of its power spectrum at a saturating Ca²⁺ concentration suggests that the macroscopic phenotype of I641A originates from an increase in transition efficacy in the initial and the final steps (manifested in the increase in $L_{32}$ and $L_{10}$) that correspond to α6 rearrangement and pore opening (Fig. 3d–f), suggesting a destabilization of the gate during these two transitions. In contrast, no substantial change in the respective equilibrium constants was observed for I550A and I551A (Fig. 3d–f), suggesting that these two residues may affect all of the connected states to a similar extent.

In the absence of Ca²⁺, the corresponding power spectra of gate mutants reveal comparable transitions as observed at a saturating Ca²⁺ concentration (Supplementary Fig. 8), which

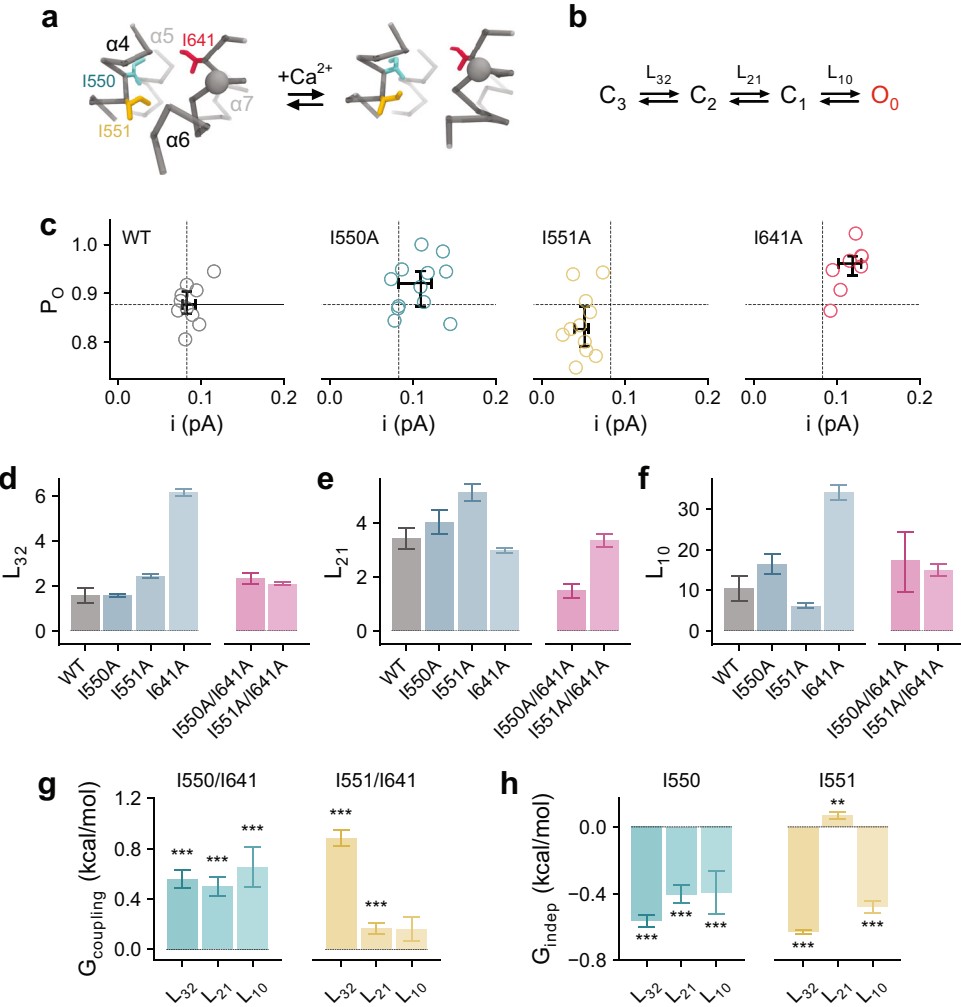

**Fig. 3 Mechanistic double-mutant cycle analysis of residues of the gate region. a** Conformational rearrangement of the inner pore entrance upon $Ca^{2+}$ binding. Cα-trace of indicated transmembrane helices of the $Ca^{2+}$-free (PDB: 5OYG) and the $Ca^{2+}$-bound conformation of TMEM16A (PDB: 5OYB) are shown. Sidechains of residues of the gate are displayed as sticks, sphere indicates position of the gating hinge. **b** Reduced mechanism depicting gating transitions at saturating $Ca^{2+}$ concentrations. **c** Single-channel current (i) and open probability (Po) estimated from non-stationary noise analysis at saturating $Ca^{2+}$ concentrations. Circles correspond to individual data points. Bars refer to median and interquartile range of data. **d–f** Equilibrium constants for mutants in the gate region for transitions **d**, $L_{32}$, **e**, $L_{21}$, and **f**, $L_{10}$. Bars indicate the best-fit values of the averaged data shown in Supplementary Fig. 7d (WT, $n = 7$; I550A, $n = 6$; I551A, $n = 7$; I641A, $n = 7$; I550A/I641A, $n = 7$; I551A/I641A, $n = 6$). Errors are 95% confidence intervals. **g** Coupling energy ($G_{coupling}$) and **h** independent energetic contribution ($G_{indep}$) of the indicated residues. Bars indicate quantities calculated using **g**, Eqs. 31–32, 35 and **h**, Eq. 34 from the best-fit values shown in **d–f**. Errors correspond to standard errors. Asterisks indicate significant deviation from zero in a two-sided one-sample *t*-test (**g** I550/I641: ***$p = 2e−8$, ***$p = 3e−7$, and ***$p = 4e−4$; I551/I641: ***$p = 1e−13$ and ***$p = 0.001$; **h** I550: ***$p = 2e−15$, ***$p = 6e−8$, and ***$p = 0.005$; I551: ***$p$-0, **$p = 0.006$, and ***$p = 2e−13$).

indicates the sampling of states that are for energetic reasons not populated in the apo state of WT. The presence of three Lorentzian components suggests a similar accessibility of states under both limiting conditions (Supplementary Fig. 8d). Although much higher than for WT, both the efficacy and the kinetics of the pre-opening ($L_{21}$) and the opening steps ($L_{10}$) are consistently lower in the apo than in the fully $Ca^{2+}$-bound state, leading to a moderate reduction in the $P_o^{max}$ (Supplementary Tables 2 and 3). This observation further confirms the role of the bound $Ca^{2+}$ ions in influencing the energetics of the gating transitions even in mutants with considerable basal activity. This enhancement acts concomitantly with the release of an electrostatic gate that impedes anion conduction in the open state of the apo channel[12].

To examine how the truncation of sidechains of Ile 550 and Ile 551 influences gating, we analyzed the respective mutations in double-mutant cycles[29,32]. In this analysis, interdependent

contributions of a mutation originating from interactions with another residue can be quantified by their coupling energy ($G_{coupling}$). $G_{coupling}$ denotes the difference between the energetic effects of a mutation introduced on the wild-type protein and in the background of the other mutation. If the two perturbations are independent, its magnitude is zero as the respective backgrounds do not have an impact on the effect of the mutation. In contrast, $G_{coupling}$ would deviate from zero if both residues interact functionally. Conversely, energetic contributions of a mutation that are independent of a particular pairwise interaction (expressed as $G_{indep}$) may be inferred from the effect of the same mutation introduced on the background of a second mutation where the sidechain of the interacting residue is truncated (see "Methods").

Given the above considerations, the positive coupling energy between Ile 550 and Ile 641 for all three gating steps, and between Ile 551 and Ile 641 where the coupling is most pronounced for the

transition $L_{32}$ (Fig. 3g), suggests that these pairs of residues functionally interact to stabilize the closed gate, an observation consistent with the macroscopic analysis that we described in an accompanying manuscript[13]. Introduction of either I550A or I551A on the background of I641A profoundly reverses the efficacy increase caused by I641A (Fig. 3d–f). For I550A this affects all gating steps to a similar extent and for I551A, the transitions $L_{32}$ and $L_{10}$ (Fig. 3h). These results indicate that while Ile 550 and Ile 551 both interact with Ile 641 to close the gate, as indicated by the positive $G_{coupling}$ in the respective double-mutant cycles (Fig. 3g), these two residues also stabilize the final state of the respective transitions independently of Ile 641, which is reflected in the negative $G_{indep}$ of either residue (Fig. 3h). Together, the data point towards a progressive destabilization of the gate region during gating starting with α6 activation and proceeding until the transition of the pore into a conductive conformation.

**An exchange of interactions at the α4-α6 interface stabilizes the open state.** In the Ile 550/Ile 641 and Ile 551/Ile 641 mutant cycles, the reversal of the I641A effect on the opening step by mutations I550A and I551A suggests that, besides their role in the stabilization of the closed gate by interacting with Ile 641, the two residues on α4 may also be involved in a stabilization of the open state by interactions that are independent of Ile 641. Because Ile 551 and Gln 649 are closely apposed in the Ca²⁺-bound structure (Fig. 4a), we investigated their potential interaction in stabilizing the open pore in a double-mutant cycle. Since the mutation Q649A consists of perturbations that change both the polarity and volume of the sidechain, we paired I551A with either Q649A or Q649L to disentangle the two effects (Fig. 4).

When introduced on its own, the mutation of Gln 649 to Ala exerts little effect on the gating transitions whereas its mutation to Leu increases the efficacy of the first and last step (Fig. 4b–d). The introduction of I551A on the Q649A background reverses the dampening effect of I551A on the opening step ($L_{10}$), which now increases the efficacy of opening (Fig. 4d). This results in a considerable negative coupling energy (Fig. 4e), suggesting that Ile 551 and Gln 649 interact to stabilize the open pore, a notion that is also supported by elevated conduction barriers in these mutants (presumably by a more collapsed pore geometry) as reflected in the outward rectification of current (see below). Examination of the stepwise mutant cycles reveals that the coupling persists when the partial charges of Gln 649 were removed on the Q649L background (Fig. 4e), suggesting that the interactions are predominantly mediated by the volume instead of the polarity of the sidechain and the open pore may thus in part be stabilized by van der Waals forces.

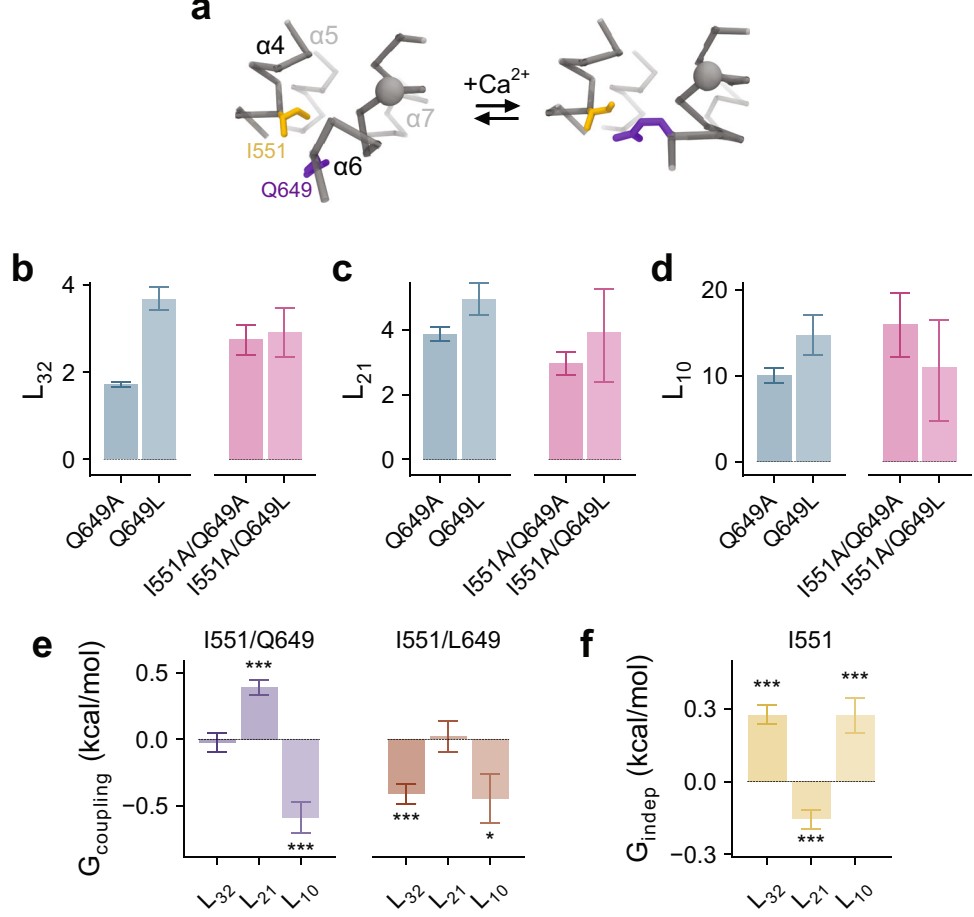

**Fig. 4 Mechanistic double-mutant cycle at the α4–α6 interface. a** Relation between Ile 551 and Gln 649 in Ca²⁺-free and -bound conformations. Representation is as in Fig. 3a. **b–d** Equilibrium constants for mutants at the interface for transitions **b**, $L_{32}$, **c**, $L_{21}$, and **d**, $L_{10}$. Bars indicate the best-fit values of the averaged data shown in Supplementary Fig. 7d (Q649A, $n = 7$; Q649L, $n = 7$; I551A/Q649A, $n = 8$; I551A/Q649L, $n = 7$). Errors are 95% confidence intervals. **e** Coupling energy ($G_{coupling}$) and **f** independent energetic contribution ($G_{indep}$) of the indicated residues. Bars indicate quantities calculated using **e**, Eqs. 31–32, 35 and **f**, Eq. 34 from the best-fit values shown in **b–d**. Errors correspond to standard errors. Asterisks indicate significant deviation from zero in a two-sided one-sample t-test (**e** I551/Q649: ***$p = 2e-7$ and ***$p = 3e-5$; I551/L649: ***$p = 5e-6$ and *$p = 0.023$; **f** ***$p = 1e-7$, ***$p = 6e-4$, and ***$p = 0.001$).

Consistent with the previously described weak interaction between Ile 551 and Ile 641 that stabilizes the closed pore in the final gating step ($L_{10}$, Fig. 3g), we show here that Ile 551 stabilizes the closed pore independently of Gln 649 (as reflected in the positive $G_{indep}$ for the same step for I551A, Fig. 4f). Together, these observations explain how I551A reverses the effect of I641A on pore opening ($L_{10}$) in the double mutant. In the described process, Gln 649 competes with Ile 641 for interaction with Ile 551, with the former interaction stabilizing the open ($O_0$) and the latter the pre-open state ($C_1$). This suggests a mechanism where Ile 551 dissociates from Ile 641 and in turn interacts with Gln 649 as the gate opens.

**Arrangement of the gate in the open pore.** Next, we investigated the interactions of the same residues when the pore is open. For quantification, we extracted the energetic effects of alanine

mutants of the three gating residues and Q649A on ion conduction and casted these in double-mutant cycles (Fig. 5a). Energies were obtained from a fit of I-V relations to a three-barrier model that was introduced previously[7,12]. The fit yields two kinetic parameters ($\sigma_\beta$ and $\sigma_h$) that can each be converted into an energy difference of the inner and central barrier relative to the outer barrier (Fig. 5b), which allows the approximate localization of effects of mutations on the ion conduction path.

Our data indicate a slight enhancement of the relative rates of anion diffusion at the inner entrance of the neck ($\sigma_\beta$) and inside the narrow pore ($\sigma_h$) for I550A and I641A due to a decrease of both barriers (Fig. 5c, d). This behavior is also reflected in the increase of the single-channel conductance in both mutants as observed in non-stationary noise analysis (Fig. 3c and Supplementary Fig. 6). Reversed effects are found in mutants Q649A and I551A, where the relative rate across the inner barrier

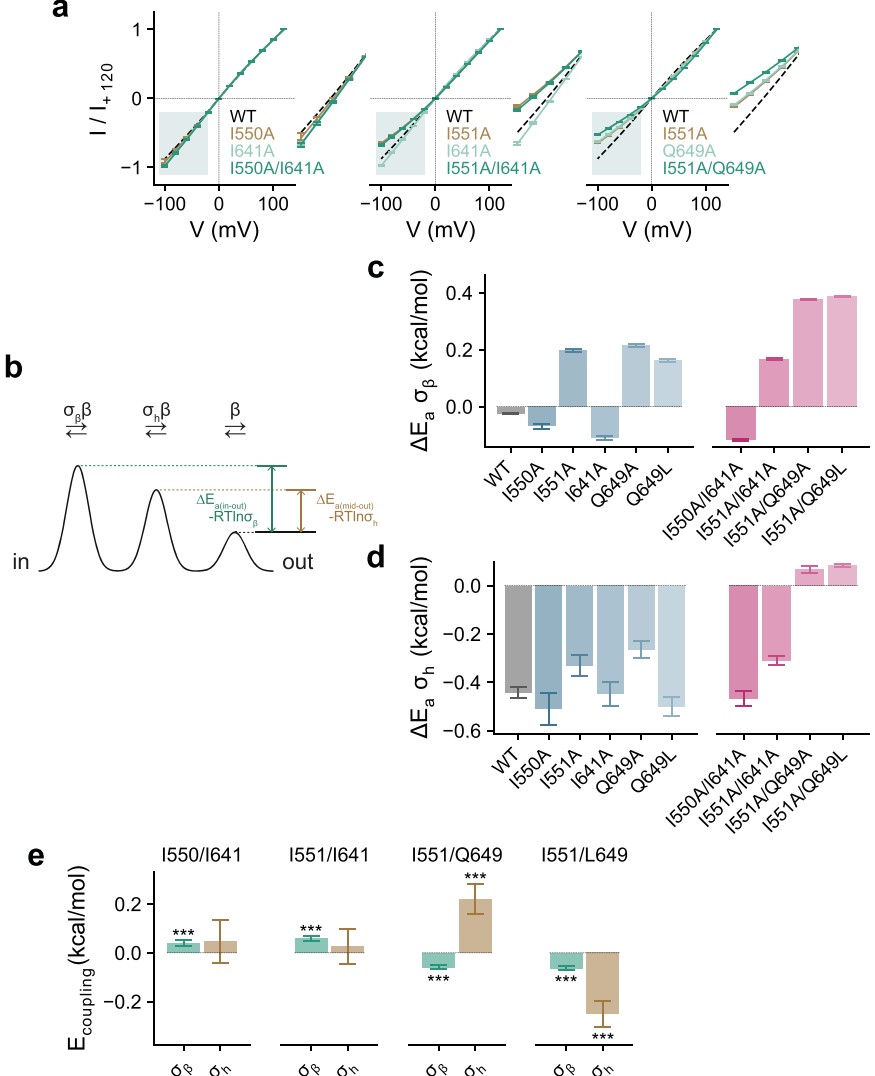

**Fig. 5 Functional coupling in the open state. a** Instantaneous I-V relations for the indicated mutants recorded at saturating $Ca^{2+}$ concentrations. Solid lines are fits to a phenomenological ion permeation model described in Eq. 1. Dashed line indicates the relation of WT. Data are averages of the indicated number of patches (WT, $n = 6$; I550A, $n = 6$; I551A, $n = 8$; I641A, $n = 7$; Q649A, $n = 6$; I550A/I641A, $n = 10$; I551A/I641A, $n = 10$; I551A/Q649A, $n = 16$), errors are SEM. **b** Minimal ion permeation model to account for the I-V relations. **c, d** Conduction parameters **c**, $\Delta E_a(\sigma_\beta)$ and **d**, $\Delta E_a(\sigma_h)$ corresponding to the barrier heights for ion conduction at the inner pore entrance and the narrow pore relative to that of the outer barrier of the indicated mutants. Bars indicate quantities calculated using Eq. 2 from the best-fit values of the averaged data shown in **a**. Errors are 95% confidence intervals. **e** Coupling energy ($E_{coupling}$) for residue pairs. Bars indicate quantities calculated using Eqs. 31–32, 35 from the values shown in **c** and **d**. Errors correspond to standard errors. Asterisks indicate significant deviation from zero in a two-sided one-sample $t$-test (I550/I641: ***$p = 0.002$; I551/I641: ***$p = 1e-8$; I551/Q649: ***$p = 4e-11$ and ***$p = 4e-4$; I551/L649: ***$p = 2e-16$ and ***$p = 6e-6$).

and the central barrier and the single-channel conductance are both decreased (Figs. 3c, 5c, d and Supplementary Fig. 6). For the I550A/I641A and I551A/I641A pairs, the coupling energy for the inner barrier is small and positive, while that for the middle barrier is not significantly different from zero (Fig. 5e). The low amplitudes of the coupling energies are consistent with a widening of the gate region in the open pore where obstruction to conduction is minimal. For the I551A/Q649A pair, however, the coupling energy is slightly negative for the inner and strongly positive for the central barrier (Fig. 5e), suggesting that both residues interact in the open state to keep the neck in a conductive conformation. Examination of the stepwise mutant cycle on the Q649L background, which removes the effects of partial charges and which consists of only contributions from steric interactions, reveals considerable negative coupling energy for the central barrier (Fig. 5e). This indicates that the positive polarity of the coupling in the overall cycle is likely due to a compensatory effect in the I551A/Q649L cycle. Therefore, while the coupling between Ile 551 and Gln 649 is small in stabilizing the open pore conformation at the inner entrance of the neck, it is required to maintain a widened conformation inside the neck. This observation is consistent with an interaction between these two residues that stabilizes the open state as inferred from kinetic analysis (Fig. 4).

**Timing of motion of the gate region**. Our analysis revealed the successive rearrangements of the gate region leading to pore opening during channel activation. To gain further insight into the location of energy maxima connecting the different states, we analyzed the gating kinetics of a series of mutants in the gate region collectively to obtain the relationship between the forward ($k_f$: $k_{32}$, $k_{21}$, $k_{10}$) or backward rate constants ($k_b$: $k_{23}$, $k_{12}$, $k_{01}$) and their corresponding forward equilibrium constants ($K_{eq}$: $L_{32}$, $L_{21}$, $L_{10}$) (Fig. 6). The relative timing of a local conformational change in a global transition can be characterized by its phi value (0–1), which is reflected in the slope when $k_f$ is plotted against $K_{eq}$ on a logarithmic scale[33–35]. A phi value close to one corresponds to a

case where a mutation affects the transition state and the final state to a similar extent. This scenario is consistent with the involvement of the site of mutation in a structural rearrangement that occurs early on the reaction coordinate. Conversely, a value close to zero would suggest a rearrangement that occurs relatively late during the global transition. During the gating of TMEM16A, the locations of respective transition states of the gate region differ for distinct transitions. Whereas both $k_f$ and $K_{eq}$ are affected to a similar extent in the series of mutants for the intermediate transition ($L_{21}$), as reflected in a phi value close to 1, $k_f$ is virtually insensitive to changes in $K_{eq}$ for transitions involving the destabilization of the gate region ($L_{32}$ and $L_{10}$) which translates to a phi value of zero (Fig. 6). In all cases, the spread of $K_{eq}$ values obtained for different mutants further emphasizes the energetic involvement of the gate region in each of the global transitions. These results suggest that during transitions in which the gate is destabilized (i.e., $L_{32}$ and $L_{10}$), the associated rearrangement occurs late and is likely amongst the final motions involved in the widening of the pore.

## Discussion

In this study, we have used electrophysiology in combination with a detailed kinetic analysis to unravel the gating mechanism of TMEM16A. By employing stationary noise analysis of macroscopic recordings at increasing $Ca^{2+}$ concentrations, we have identified distinct states of the channel that are successively occupied during activation. Transitions that dominate at saturating $Ca^{2+}$ concentrations where binding events are scarce are consistent with the sampling of at least four distinct protein conformations (Fig. 7a). The strongly $Ca^{2+}$-dependent step corresponds to an early transition where the helix α6 is either loose as defined in the structure of TMEM16A obtained under $Ca^{2+}$-free conditions or tight as observed in the $Ca^{2+}$-bound form of the protein. The conformational rearrangement of α6 is coupled to the gate at the intracellular entry to the narrow neck, and leads to its successive destabilization via another pre-open intermediate to finally reach a conductive conformation.

Our kinetic analysis suggests that successive $Ca^{2+}$ binding to TMEM16A proceeds via distinct trajectories depending on its concentration (Fig. 7a). Whereas the first $Ca^{2+}$ binds to a state where α6 is in its loose conformation, the binding of the second $Ca^{2+}$ would proceed in the same α6-loose conformation only at unphysiologically high $Ca^{2+}$ concentrations, while it would occur more frequently in the state where α6 has assumed an activated conformation at physiological $Ca^{2+}$ concentrations. The mutual stabilization of the two events increases the affinity for the binding of the second $Ca^{2+}$, underlying the positive cooperativity of channel activation.

Following the binding of two $Ca^{2+}$ ions and α6 activation, the protein further progresses towards pore opening. The presence of pre-open intermediates during this transition suggests that the channel undergoes successive conformational rearrangements before entering a conducting state. This could be reflected in the $Ca^{2+}$-bound structure of TMEM16A, where such a pre-open intermediate might have been stabilized in a detergent environment and the absence of bound $PI(4,5)P_2$, the latter of which prevents current rundown in excised patch recordings[36,37]. In this structure, α6 has already undergone its ligand-induced rearrangement whereas the narrowest part of the pore might not have expanded sufficiently to accommodate permeant anions. Although still not conductive, the partial widening of the inner pore in this structure signifies a destabilization of the gate region as a likely consequence of the tightened conformation of α6. Such coupled movement is also manifested in our kinetic analysis where Ile 641 was found to stabilize the relaxed conformation of

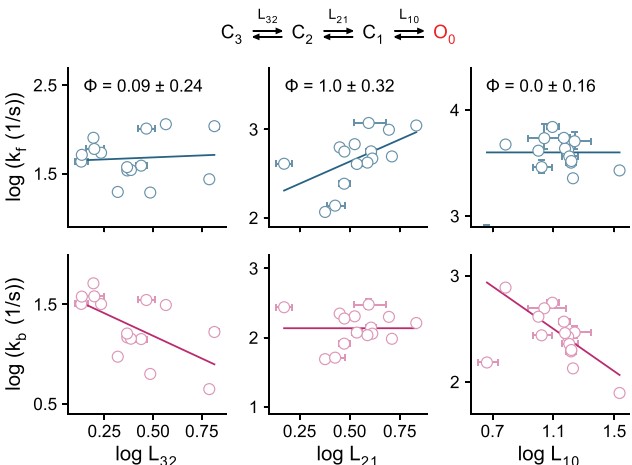

$$C_3 \underset{}{\overset{L_{32}}{\rightleftarrows}} C_2 \underset{}{\overset{L_{21}}{\rightleftarrows}} C_1 \underset{}{\overset{L_{10}}{\rightleftarrows}} O_0$$

**Fig. 6 Rate-equilibrium free-energy relations of the gating steps.** Plots show the experimental relation between the forward (blue) and backward (red) rate constants and the corresponding equilibrium constants for the transitions relating the gating intermediates and end points (from $C_3$ to $O_0$). Each data point corresponds to the parameter(s) obtained for a mutant in the gate region shown in Supplementary Fig. 7d. Errors correspond to 95% confidence intervals (respective error bars are not visible when they are smaller than the symbols). Solid lines are fits to the pair of rate-equilibrium free energy relations for each transition (Eq. 36). The estimated value of phi is presented as best-fit ± standard error.

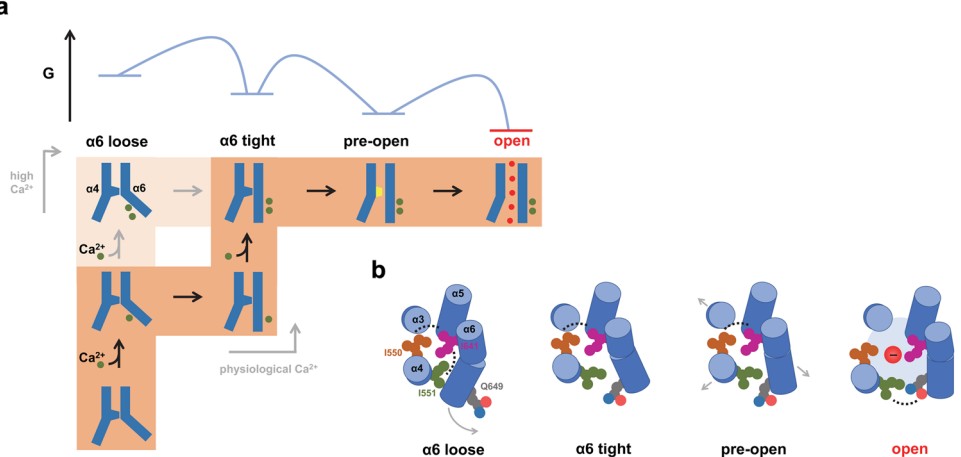

**Fig. 7 Activation mechanism. a** Cartoon depiction of the activation mechanism. The states are as in Fig. 1d. Inset (top) shows a schematic representation of the energy maxima in the gate region between connected states. The relative location of the transition states was obtained from phi value analysis. The two alternative pathways for activation at physiological and at high $Ca^{2+}$ concentrations are indicated. **b** Schematic depiction of the relationship between residues of the gate region in different steps during activation. The view is from the extracellular side. Gate residues located on α4 (Ile 550 and Ile 551) retain their interaction with the gate residue Ile 641 on α6 in the transition between the three closed conformations although the interaction between Ile 551 and Ile 641 successively weakens until it is replaced by an interaction with Gln 649 upon transition into the open state where the gate has dissociated and the pore has dilated to permit ion conduction.

α6 (α6-loose) by coupling with Ile 550 and Ile 551 in the gate region (Fig. 3). Consistent with this assumption, the disruption of any of the three gate residues in the apo state is likely to have a reciprocal energetic consequence on the α6 conformation, as observed in our kinetic model and also in the cryo-EM structure of the activating mutant I551A described in our accompanying study[13].

The spectra of mutants with basal activity obtained in the absence of $Ca^{2+}$ reveal constitutive sampling of equivalent transitions as found in the $Ca^{2+}$-bound state, underlining that the general mechanism of pore opening does not strictly require the presence of the ligand as expected for a mechanism that is based on an MWC model[27,28]. Despite the similarity of the accessible states, local structural differences of the open states in the presence and absence of $Ca^{2+}$ might be expected given the strong electrostatic influence of the bound $Ca^{2+}$ ions on the protein conformation. Such differences have been observed in the apo structure of I551A described in an accompanying study[13], where α6 adopts a partially activated conformation in which the transition from an α- to a π-helix conformation cannot be completed due to the energetic penalty of the transition and the electrostatic repulsion from the vacant binding site. Thus, although activation and pore opening can proceed without a transition into the strained π-helix conformation, the decrease in the efficacy of both the pre-opening ($L_{21}$) and opening steps ($L_{10}$) in the apo state suggests that this might have an energetic consequence on gating.

An opening equilibrium constant in WT that is several fold larger than one suggests that the partially destabilized gate found in the α6-tight and pre-open intermediates is intrinsically less stable compared to the fully open conformation (Fig. 1d). This might be attributable to the combined effect of weakened inter-residue interactions in the destabilized state and the release of ordered water around the hydrophobic cage in the dewetted region upon opening. On the basis of coupling energies obtained from mutant cycles (Figs. 3–6), we come up with a structural interpretation of the events leading to the opening of the gate (Fig. 7b). Immediately preceding pore opening, retained interactions between Ile 550 and 551 on α4 with Ile 641 on the opposing α6 still stabilize the partially widened gate region, thereby maintaining a pore diameter that is sufficiently small to favor spontaneous dewetting, a condition prohibitive for ion conduction. Pore opening results as these interactions become disrupted and is accompanied by the formation of an alternative arrangement between α4 and α6 where Ile 551 engages in an interaction with the neighboring Gln 649 (on α6) to stabilize the open pore. This sequence of events is compatible with the at times counterintuitive characteristics of Ile 551, which, by toggling respectively between interaction with Ile 641 or Gln 649 on α6, may act as a switch that stabilizes either the pre-open or the open pore conformations.

Further insight into the relative location of transition states connecting the gating intermediates and end states was obtained from phi value analysis, which relates changes in the rate and corresponding equilibrium constants. Whereas a phi value close to one in the passage from the 'α6-tight' to the 'pre-open' conformation suggests a transition state that is reached early on the reaction coordinate, phi values close to zero for the first and last gating steps are consistent with rate-limiting steps towards the end of the process (Fig. 7a). For the last step, this suggests that the rearrangement of the gate is amongst the final events in a delocalized motion that opens the pore (Fig. 7b). Thus, although the gate region remains in a closed-like conformation until the end of the final gating step, other parts surrounding the pore might have already approached an open-like state earlier in the gating process. In this respect, a more extended rearrangement elsewhere in the pore region is likely, as evident from systematic mutagenesis that we report in an accompanying manuscript[13] where residues located towards the extracellular end of the neck also impact gating and contribute to a widening of the central part of the pore in the open conformation.

In conclusion, by establishing a general method to measure steady-state kinetics for channels whose low unitary conductance prohibits single-channel recording, we have provided a first detailed mechanistic view of the gating process in TMEM16A. While our current study has focused on the gate region, the expansion of the mutational analysis towards other residues of the pore might further define the molecular motions during gating. Given the conserved architecture of the family, our findings will be relevant for studying activation in other TMEM16 ion channels and lipid scramblases, and provide a functional template

for the therapeutic targeting of TMEM16A in pathological conditions such as cystic fibrosis[38].

## Methods

**Molecular biology and cell culture**. HEK293T cells (ATCC CRL-1573) were maintained in Dulbecco's modified Eagle's medium (DMEM; Sigma-Aldrich) supplemented with $10\,U\,ml^{-1}$ penicillin, $0.1\,mg\,ml^{-1}$ streptomycin (Sigma-Aldrich), 2 mM L-glutamine (Sigma-Aldrich), and 10% FBS (Sigma-Aldrich) in a humidified atmosphere containing 5% $CO_2$ at 37 °C. HEK293T cells were transfected with 3 µg DNA per 6 cm Petri dish using the calcium phosphate co-precipitation method and were used within 24–96 h after transfection. Mutations were introduced with a modified QuikChange method[39] using the $a,c$ variant of mouse TMEM16A (UniProt identifier: Q8BHY3-1) as the template and were verified by sequencing. Primers are listed in Supplementary Table 4.

**Electrophysiology**. Recordings were performed on inside-out patches excised from HEK293T cells expressing the construct of interest. Transfected cells were identified via the fluorescence of the Venus tag. Patch pipettes were pulled from borosilicate glass capillaries (O.D. 1.5 mm, I.D. 0.86 mm; Sutter Instrument) and were fire-polished with a microforge (Narishige) before use. Pipette resistance was typically 3–8 MΩ when filled with the recording solutions detailed below. Seal resistance was typically 4 GΩ or higher. Voltage-clamp recordings were made using Axopatch 200B, Digidata 1550, and Clampex 10.6 (Molecular Devices). Analog signals were filtered with the in-built 4-pole Bessel filter at 10 kHz and were digitized at 20 kHz. Solution exchange was achieved using a gravity-fed system through a theta glass pipette mounted on an ultra-fast piezo-driven stepper (Siskiyou). Liquid junction potential was found to be consistently negligible given the ionic composition of the solutions and was therefore not corrected. All recordings were performed at 20 °C.

A symmetrical ionic condition was used throughout. Stock solution with $Ca^{2+}$-EGTA contained 150 mM NaCl, 5.99 mM Ca(OH)$_2$, 5 mM EGTA, and 10 mM HEPES at pH 7.40. Stock solution with EGTA contained 150 mM NaCl, 5 mM EGTA, and 10 mM HEPES at pH 7.40. Free $Ca^{2+}$ concentrations were adjusted by mixing the stock solutions at the required ratios calculated using the WEBMAXC program (http://web.stanford.edu/~cpatton/webmaxcS.htm). Patch pipettes were filled with the stock solution with $Ca^{2+}$-EGTA, which has a free $Ca^{2+}$ concentration of 1 mM.

**Analysis of current-voltage (I-V) relations**. I-V data were fitted to a minimal permeation model that accounts for the most fundamental biophysical behavior of mTMEM16A as described previously[7,12],

$$I = zFAe^{\frac{zFV}{2nRT}} \frac{c_i - c_o e^{-\frac{zFV}{RT}}}{e^{-zFV\frac{n-1}{nRT}} + \left(\frac{1}{\sigma_h}\right)\frac{1-e^{-zFV\frac{n-2}{nRT}}}{e^{\frac{zFV}{nRT}}-1} + \frac{1}{\sigma_\beta}} \quad (1)$$

where $I$ is the current, $n$ is the number of barriers, $c_i$ and $c_o$ are the intracellular and extracellular concentrations of the charge carrier, $z$ is the valence of $Cl^-$, $V$ is the membrane voltage, and $R$, $T$, and $F$ have their usual thermodynamic meanings. $A = \beta_0 v$ is a proportionality factor where $\beta_0$ is the value of $\beta$ when $V = 0$ and $v$ is a proportionality coefficient that has a dimension of volume. $\sigma_h$ and $\sigma_\beta$ are respectively the rate of barrier crossing at the middle and the innermost barriers relative to that at the outermost barrier ($\beta$). The best-fit values of $\sigma_\beta$ and $\sigma_h$ at zero and saturating $Ca^{2+}$ concentrations were used to calculate $\Delta E_{a(\sigma\beta)}$ and $\Delta E_{a(\sigma h)}$, the difference between the activation energy at the innermost barrier and the middle barrier relative to that of the outermost respectively, using

$$\Delta E_{a(\sigma_\beta)} = -RT \ln \sigma_\beta$$
$$\Delta E_{a(\sigma_h)} = -RT \ln \sigma_h \quad (2)$$

**Non-stationary noise analysis**. Non-stationary noise analysis was performed as described previously[10,40]. The current and variance were sampled by repeatedly activating and deactivating the channel using regularly spaced concentration jumps. The variance of such 50–100 aligned successive and kinetically identical currents at each time point during deactivation was calculated by computing the mean of the squared successive difference[41], which mitigates the effect of non-stationarity at each isochrone and therefore allows the estimation of the variance in the presence of current rundown. Assuming the presence of a dominating conducting level, the data were fitted to

$$\sigma^2_{total} = \sigma^2_N + \sigma^2_{bg}$$
$$\sigma^2_N = i(\bar{I} - \bar{I}_{bg}) - \frac{(\bar{I} - \bar{I}_{bg})^2}{N} \quad (3)$$

where $\sigma^2_N$ is the variance for $N$ channels, $i$ is the unitary current, $\bar{I}$ is the mean current, and the subscript bg denotes background. The introduction of $\bar{I}_{bg}$ and $\sigma^2_{bg}$ allows the accommodation of x- and y- translations respectively. To merge data from different patches, individual $\sigma^2_{bg}$- and $\bar{I}_{bg}$-corrected $\sigma^2_N$-$\bar{I}$ plots were brought to the same scale by normalization in both x and y directions according to

a patch-specific parameter $iN$, the maximum achievable $\bar{I}$ for each patch if the $Po$ was 1. Because each data pair ($\bar{I}_j, \sigma^2_{N_j}$) is unique, they were sorted according to the $\bar{I}$ values and were averaged using a Gaussian moving average filter, which allowed the central tendency of the normalized parabolas from different patches to be estimated. The averaged data were re-fitted to Eq. 3 without the $\sigma^2_{bg}$ and $\bar{I}_{bg}$ terms. This procedure allows the estimated $Po$ to be directly read from the merged $\sigma^2_N$-$\bar{I}$ plots.

**Computing the power spectrum**. The power spectrum of a steady-state current recorded at +80 mV was obtained via Fast Fourier transform (FFT). For analysis, currents were recorded over a continuous period of typically 50 s, or 100 s for cases where sampling of lower frequencies was required. Recordings were filtered at 10 kHz and sampled at 20 kHz as described above. To reduce spectral leakage, a Hamming window was applied to mitigate edge discontinuities before Fourier transform[42]. The magnitude of the spectral components ($S$) is given by the sum of squares of the amplitudes derived from the real ($a_{real}$) and imaginary parts ($a_{imag}$),

$$S = a^2_{real} + a^2_{imag} \quad (4)$$

which was scaled according to Clampfit 10.6 (Molecular devices) using

$$P = S\frac{2N}{f\bar{\omega}}$$
$$\bar{\omega} = \frac{\sum_{i=0}^{N-1} f^2_{w_i}}{N} \quad (5)$$

to yield $P$, the one-sided spectral density in $\frac{A^2}{Hz}$, where $N$ is the transform length, $f$ is the sampling frequency in Hz, and $\bar{\omega}$ is a scale factor for the window function ($f_w$).

The background spectrum of each patch, recorded at 0 mV where the current reverses, was subtracted from the raw spectrum to obtain the power spectrum specific to the channel. This resulting spectrum typically consists of three major components, which are the 1/f-like component at low frequency[18,43], the Lorentzian components corresponding to molecular fluctuations, and an effectively constant term likely consisting of very fast components whose corner frequencies are not resolved within the bandwidth of the spectrum. In order to subtract the 1/f-like and constant components, the total spectrum was fitted to

$$P = \frac{a_0}{f^n} + \sum_i a_i \frac{1}{1 + \left(f/f_{c_i}\right)^2} + c \quad (6)$$

to obtain an empirical description, where $f$ is frequency, $n$ is an exponent describing the decay, $a_0$ and $a_i$ are respectively the amplitude of the 1/f-like and the Lorentzian components, $f_{ci}$ is the corner frequency, and $c$ is a constant. The number of components was assessed using the AIC calculated for models with different number of transitions[26]. It was found that at least three Lorentzian components were required to account for the spectrum for cases at saturating $Ca^{2+}$ concentrations.

**Mechanism and parameter estimation**. We hypothesized that TMEM16A gating can be described by the following mechanism,

$$C_3 \leftrightarrow C_2 \leftrightarrow C_1 \leftrightarrow O_0$$

where C and O correspond to the closed and open states respectively, and the subscripts denote the number assigned to the states. The matrix notation of this mechanism[44] is

$$\mathbf{Q} = \begin{bmatrix} -k_{01} & k_{01} & 0 & 0 \\ k_{10} & -k_{10}-k_{12} & k_{12} & 0 \\ 0 & k_{21} & -k_{21}-k_{23} & k_{23} \\ 0 & 0 & k_{32} & -k_{32} \end{bmatrix} \quad (7)$$

where the subscripts indicate the transition described by the rate constant $k$ in $s^{-1}$, for example, $k_{01}$ corresponds to the rate constant of the transition from state 0 to 1. In the case where the $Ca^{2+}$ binding steps were included, the following mechanism, with the superscripts denoting the number of $Ca^{2+}$ bound, was used

$$\begin{array}{ccccccc} C_3^{x_2} & \leftrightarrow & C_2^{x_2} & \leftrightarrow & C_1^{x_1} & \leftrightarrow & O_0^{x_2} \\ \updownarrow & & \updownarrow & & & & \\ C_5^{x} & \leftrightarrow & C_4^{x} & & & & \\ \updownarrow & & & & & & \\ C_6 & & & & & & \end{array}$$

and the corresponding matrix is given by

$$
\mathbf{Q} = \begin{bmatrix}
-k_{01} & k_{01} & 0 & 0 & 0 & 0 & 0 \\
k_{10} & -k_{10}-k_{12} & k_{12} & 0 & 0 & 0 & 0 \\
0 & k_{21} & -k_{21}-k_{23}-k_{24} & k_{23} & k_{24} & 0 & 0 \\
0 & 0 & k_{32} & -k_{32}-k_{35} & 0 & k_{35} & 0 \\
0 & 0 & xk_{42} & 0 & -xk_{42}-k_{45} & k_{45} & 0 \\
0 & 0 & 0 & xk_{53} & k_{54} & -xk_{53}-k_{54}-k_{56} & k_{56} \\
0 & 0 & 0 & 0 & 0 & xk_{65} & -xk_{65}
\end{bmatrix}
\tag{8}
$$

where $x$ is the molar concentration of $Ca^{2+}$. In both cases, the below relation was used to decrease the number of free parameters

$$
k_{10} = k_{01}\left(\frac{P_O}{1-P_O}\right)\left(\frac{1+L_{32}+L_{32}L_{21}}{L_{32}L_{21}}\right)
\tag{9}
$$

where $L$ is the forward equilibrium constant with the subscript indicating the transition, and $P_o$ was supplied as an experimental estimate from non-stationary noise analysis at saturating $Ca^{2+}$ concentrations ($P_{OCa\to\infty}$). At zero $Ca^{2+}$, the open probability ($P_{O0Ca}$) was calculated by rearranging

$$
\frac{\bar{I}_{0Ca}}{\bar{I}_{Ca\to\infty}} = \frac{i_{0Ca}P_{O0Ca}}{i_{Ca\to\infty}P_{OCa\to\infty}}
\tag{10}
$$

where $\bar{I}$ is the mean current, $i$ the unitary current, and the subscripts indicate at zero $Ca^{2+}$ (0Ca) and at saturation (Ca → ∞), which were derived from experimental values. In the case of the full mechanism, microscopic reversibility[45] and the knowledge of the highest $Ca^{2+}$ binding affinity ($K_{d(a2)}$), which was empirically estimated to be $3.6 \times 10^{-8}$ M in an accompanying manuscript[13], were also used

$$
k_{45} = \frac{k_{23}k_{35}k_{54}}{K_{d(a2)}k_{53}k_{32}}
$$
$$
k_{24} = k_{42}K_{d(a2)}
\tag{11}
$$

and the on rate of the first $Ca^{2+}$ binding step was assigned a diffusion-limited value $(1 \times 10^{11})$[46].

Following the methods of Colquhoun and Hawkes[44], the equilibrium occupancy of states was calculated from

$$
\mathbf{P}(\infty) = \mathbf{Y}_0\left(\mathbf{V}_{\lambda=0}\mathbf{V}_{\lambda=0}^{-1}\right)
\tag{12}
$$

where $\mathbf{Y}_0$ is the initial occupancy and $\mathbf{V}$ was obtained from the Eigen decomposition of $\mathbf{Q}$

$$
\mathbf{Q} = \mathbf{V}\mathbf{\Lambda}\mathbf{V}^{-1}
\tag{13}
$$

and

$$
\mathbf{\Lambda} = \begin{bmatrix} \lambda_1 & & \\ & \ddots & \\ & & \lambda_n \end{bmatrix}
$$
$$
\mathbf{V} = \begin{bmatrix} v_{11} & \cdots & v_{n1} \\ \vdots & \ddots & \vdots \\ v_{1n} & \cdots & v_{nn} \end{bmatrix}
\tag{14}
$$

are the Eigenvalue and Eigenvector matrices respectively. The corresponding spectral matrices are given by

$$
\mathbf{A}_i = \mathbf{V}_{i^{th}col}\mathbf{V}_{i^{th}row}^{-1}
\tag{15}
$$

The general form of the single-sided power spectrum due to Markovian fluctuations is given by[19]

$$
G(f) = 4NV^2\mathbf{P}_o(\infty)\mathbf{\Gamma}_o\left[\sum_{i=2}^{n}\mathbf{A}_{ioo}\frac{-\lambda_i^{-1}}{1+\left(\frac{2\pi f}{\lambda_i}\right)^2}\right]\mathbf{\Gamma}_o\mathbf{u}_o
\tag{16}
$$

where $N$ is the number of conducting units, $V$ is the membrane potential when the current is linear and reverses at 0 mV,

$$
\mathbf{P}_o(\infty) = \mathbf{P}(\infty)_{(o_1\ldots o_k)}
\tag{17}
$$

is the steady-state occupancy of open states 1 to k,

$$
\mathbf{A}_{i_{oo}} = \mathbf{A}_{i(o_1\ldots o_k, o_1\ldots o_k)}
\tag{18}
$$

is a submatrix of the spectral matrix and $o_1\ldots o_k$, $o_1\ldots o_k$ denote the upper left elements,

$$
\mathbf{\Gamma}_o = \begin{bmatrix} \gamma_{o_1} & & \\ & \ddots & \\ & & \gamma_{o_k} \end{bmatrix}
\tag{19}
$$

is the conductance of the states arranged in a matrix form, and

$$
\mathbf{u}_o = \begin{bmatrix} 1 \\ \vdots \\ 1 \end{bmatrix}
\tag{20}
$$

is a unit vector of length corresponding to the number of open states. Because the amplitude of the power spectrum concerns the number of channels and their conductance, which are variables not related to the mechanism, we fitted the experimental power spectra using a normalized form

$$
G_{norm}(f) = \frac{G(f)}{G(0)}
\tag{21}
$$

where $G(0)$ is a constant corresponding to the power at a very low frequency.

For spectra obtained at saturating concentrations, model parameters ($\theta$) were estimated by minimizing the sum of squares between Eq. 16 (4-state Q-matrix, concentration-independent) and the experimental spectra ($y(f_j)$).

$$
\epsilon_G(\theta) = \sum_j\left[G\left(f_j, \theta, P_{OCa\to\infty}\right) - y(f_j)\right]^2
\tag{22}
$$

In the case where a family of power spectra was fitted, the sum of squares between Eq. 16 (7-state Q-matrix, concentration-dependent) and the experimental spectra obtained at the indicated $Ca^{2+}$ concentrations ($y(x_i, f_j)$)

$$
\epsilon_G(x_i, \theta) = \sum_j\left[G\left(x_i, f_j, \theta, P_{OCa\to\infty}\right) - y(x_i, f_j)\right]^2
\tag{23}
$$

and those between Eq. 12 and the experimental open probability ($p(x_i)$)

$$
\epsilon_{P_{O\infty}}(x_i, \theta) = \left[P_{O\infty}(x_i, \theta) - p(x_i)\right]^2
\tag{24}
$$

were used to calculate the total sum of squares

$$
\epsilon_{total}(\theta) = \sum_i\epsilon_G(x_i, \theta) + z\sum_i\epsilon_{P_{O\infty}}(x_i, \theta)
\tag{25}
$$

where $z$ is a scaling factor. The variance of the best-fit parameters was obtained from the diagonal elements of the variance-covariance matrix[47,48]

$$
\mathbf{H}^{-1} = \left(\mathbf{J}^T\cdot\mathbf{J}\right)^{-1}
$$

multiplied by

$$
\sigma^2 = \frac{\epsilon_G(\theta) \; or \; \epsilon_{total}(\theta)}{n_d - n_p}
\tag{26}
$$

where $\mathbf{H}$ and $\mathbf{J}$ are the Hessian and Jacobian matrices at the least squares estimates respectively, the superscript $T$ indicates transpose, and $n_d$ and $n_p$ are the number of data points and parameters respectively. The square root of the variance was used to approximate the standard deviation error, from which the 95% confidence interval was computed.

**Validation of kinetic analysis via simulations**. As discussed in the results section, the power spectrum provides an alternative means to analyze microscopic kinetics for channels where single-channel recording is challenging. Although mechanistic information can be inferred, direct fitting of the power spectrum is not possible without additional experimental information as the number of rate constants to be estimated is always larger than the number of observables (amplitudes and corner frequencies of the Lorentzian components) even for linear mechanisms. This limitation may be overcome by using the open probability (Po), which can be estimated independently from non-stationary noise analysis, as an additional constraint.

To validate this approach, we analyzed the accuracy and precision of parameter estimation when the independent experimental information is combined. As described below, we randomly sampled the power spectra of simulated single-channel trajectories and Po from a Gaussian distribution with a spread similar to the experimental standard error, and estimated parameters from these synthetic data. The distributions and correlations of these estimates are shown in Supplementary Fig. 3. As expected from an adequately determined system, the 95% confidence intervals are finite and the estimates are centered at the true value of the parameter (Supplementary Fig. 3a). Moreover, although the estimates of some of the rate constants are somewhat correlated, the estimates of the equilibrium constants show little if any correlation (Supplementary Fig. 3b, c). These results indicate that both the rate and equilibrium constants estimated from the power spectrum can be interpreted meaningfully for linear mechanisms.

In addition, as most experimental records are not truly stationary, we analyzed the effect of non-stationarity and/or current rundown, typical of TMEM16A current recorded from excised patches, on simulated power spectra. In our examples, we modeled channel rundown as a stochastic first-order decay in the number of channels (Supplementary Fig. 4a) and, as an alternative mechanism, a slow entry into a non-conducting state from the open state (Supplementary Fig. 4d). In both cases, the resulting power spectrum can be described by the superposition of the spectrum describing the mechanism and that describing the

autocorrelation of the survival trajectory of the number of activatable channels (Supplementary Fig. 4b, c, e, f). This is likely a consequence of the separation of time scales, meaning that the concurrent current decay does not affect the determination of model parameters when it is slow compared to the time scale of interest and that its power can be subtracted from the experimental spectrum to yield the mechanism-dependent components.

**Analysis of the gating pathway**. Similar to Colquhoun and Lape[49], time-independent probabilities for a transition out of a specified state were calculated from the optimized model parameters according to

$$\pi_{ij} = \frac{k_{ij}}{\sum_{j\neq i} k_{ij}}, j \neq i \tag{27}$$

Specific to the mechanism that we proposed, two alternate routes of $Ca^{2+}$ binding can be taken during activation. One route consists of two consecutive binding events before the channel undergoes any gating transitions ($6\rightarrow5\rightarrow3\rightarrow2\rightarrow1\rightarrow0$), which we term non-coupled binding, and the other is coupled to a conformational change ($6\rightarrow5\rightarrow4\rightarrow2\rightarrow1\rightarrow0$), termed coupled binding. The probabilities for these routes are

$$P_{non-coupled\ binding} = \pi_{65}\pi_{53}\pi_{32}\pi_{21}\pi_{10}$$
$$P_{coupled\ binding} = \pi_{65}\pi_{54}\pi_{42}\pi_{21}\pi_{10} \tag{28}$$

and are dependent on $Ca^{2+}$ concentration. The relative usage of these routes was calculated using

$$\frac{P_{route\ i}([Ca^{2+}])}{\sum_i P_{route\ i}([Ca^{2+}])} \tag{29}$$

for the range of $Ca^{2+}$ concentrations displayed in Fig. 2d. The preferred route of deactivation was calculated as above, except that the corresponding sequence of transitions was flipped.

**Double-mutant cycle analysis**. The free energy of transition ($\Delta G$) was calculated from the forward equilibrium constant using

$$\Delta G_{ij} = -RT \ln L_{ij} \tag{30}$$

where $R$ and $T$ have their usual thermodynamic meanings, $L$ is the forward equilibrium constant and the subscript indicates the transition from state $i$ to $j$. A double-mutant cycle[29,50] can be described by the following scheme

$$
\begin{array}{ccc}
 & \Delta\Delta G_{ij}^{(0-X,Y)} & \\
X, Y & \rightarrow & 0, Y \\
\Delta\Delta G_{ij}^{(X,0-Y)}\ \downarrow & & \downarrow\ \Delta\Delta G_{ij}^{(0,0-Y)} \\
X, 0 & \rightarrow & 0, 0 \\
 & \Delta\Delta G_{ij}^{(0-X,0)} &
\end{array}
$$

where $X$ and $Y$ are two residues of interest and 0 denotes a mutation. The change in the free energy of transition ($\Delta\Delta G$) caused by a mutation was calculated as

$$\Delta\Delta G_{ij}^{(0-X,Y)} = \Delta G_{ij}^{(0,Y)} - \Delta G_{ij}^{(X,Y)}$$
$$\Delta\Delta G_{ij}^{(X,0-Y)} = \Delta G_{ij}^{(X,0)} - \Delta G_{ij}^{(X,Y)}$$
$$\Delta\Delta G_{ij}^{(0-X,0)} = \Delta G_{ij}^{(0,0)} - \Delta G_{ij}^{(X,0)}$$
$$\Delta\Delta G_{ij}^{(0,0-Y)} = \Delta G_{ij}^{(0,0)} - \Delta G_{ij}^{(0,Y)} \tag{31}$$

The coupling energy between $X$ and $Y$ ($\Delta\Delta\Delta G^{XY}$) was calculated using either the $X$ or $Y$ mutations

$$\Delta\Delta\Delta G_{ij}^{XY} = \Delta\Delta G_{ij}^{(0-X,0)} - \Delta\Delta G_{ij}^{(0-X,Y)}$$
$$= \left(\Delta G_{ij}^{(0,0)} - \Delta G_{ij}^{(X,0)}\right) - \left(\Delta G_{ij}^{(0,Y)} - \Delta G_{ij}^{(X,Y)}\right) \tag{32}$$

The perturbation caused by the mutation $X \rightarrow 0$ on the wild-type background ($X, Y$) may be thought of as a combination of effects interdependent ($\Delta G_{ij}^{XY}$) as well as independent ($\Delta G_{ij}^{X}$) of the other residue $Y$ in the cycle, hence

$$\Delta\Delta G_{ij}^{(0-X,Y)} = \Delta G_{ij}^{XY} + \Delta G_{ij}^{X} \tag{33}$$

Assuming that the inter-dependent component ($\Delta G_{ij}^{XY}$) is largely abolished when the same mutation $X \rightarrow 0$ is analyzed on a background when $Y$ is mutated ($X, 0$), this perturbation may be interpreted as

$$\Delta\Delta G_{ij}^{(0-X,0)} \approx \Delta G_{ij}^{X} \tag{34}$$

and the coupling energy as

$$\Delta\Delta\Delta G_{ij}^{XY} = \Delta\Delta G_{ij}^{(0-X,0)} - \Delta\Delta G_{ij}^{(0-X,Y)} \approx -\Delta G_{ij}^{XY} \tag{35}$$

Consequently, the interdependent component ($\Delta G_{ij}^{XY}$) might be extracted from the coupling energy ($\Delta\Delta\Delta G_{ij}^{XY}$), and the independent effect ($\Delta G_{ij}^{X}$) may be approximated by $\Delta\Delta G_{ij}^{(0-X,0)}$.

The standard error ($\sigma$) of the parameter estimates for each subtraction was propagated as described in the Statistics section. Deviation of $\Delta\Delta\Delta G_{ij}^{XY}$ from zero was detected using a two-sided one-sample t-test with a significance level of 0.05.

**Rate-equilibrium free-energy relation analysis**. The rate-equilibrium free-energy relation[34,51] consists of the following pair of relations

$$\log k_f = \log k_i + \phi \log L$$
$$\log k_b = \log k_i + (\phi - 1) \log L \tag{36}$$

that describe the effect of a series of perturbations on the rate constants ($k_f$ and $k_b$) as a fraction ($\phi$ and $\phi - 1$) of their effect on the forward equilibrium constant $L$. $\phi$ can adopt values between 0 and 1. $k_i$ is the rate constant when $L = 1$. The parameters $\phi$ and $k_i$ were estimated by minimizing the total sum of squares for the set of equations for each transition.

**Simulation of single-channel records**. Trajectories of single-channel opening and closing ($X_t$) were generated using an approximation of a stochastic simulation algorithm[52]. The initial state sampled at $t = 0$ was assigned according to the deterministic equilibrium occupancy of the states ($P_1 \ldots P_n$) (Eq. 12). A random number ($r$) was drawn from the uniform distribution in the unit interval and the following acceptance criteria were used to determine which state is sampled,

If $r < P_i$ then $X_{t=0} = $ state $i$
If $P_i \leq r < P_i + P_j$ then $X_{t=0} = $ state $j$
If $P_i + P_j \leq r < P_i + P_j + P_k$ then $X_{t=0} = $ state $k$

and so forth. Once initialized, the transition to the next state was assigned according to the transition probability in the infinitesimal time interval ($dt$)[44],

$$p_{ij}(dt) = prob\left(state_j\ at\ t + dt | state_i\ at\ t\right) \overset{\Delta}{=} \lim_{dt\rightarrow0}\left(k_{ij}dt\right) \tag{37}$$

A random number ($r$) was again drawn from the uniform distribution in the unit interval and the following acceptance criteria were used to determine which state is sampled next,

If $r < p_{ij}(dt)$ then $X_{t+dt} = $ state $j$ given that $X_t = $ state $i$
If $p_{ij}(dt) \leq r < p_{ij}(dt) + p_{ik}(dt)$ then $X_{t+dt} = $ state $k$ given that $X_t = $ state $i$
If $p_{ij}(dt) + p_{ik}(dt) \leq r < p_{ij}(dt) + p_{ik}(dt) + p_{il}(dt)$ then $X_{t+dt} = $ state $l$ given that $X_t = $ state $i$

and so forth. This procedure was then repeated until the desired length of the trajectory is reached.

The above algorithm is a finite approximation to our choice of the infinitesimal interval $dt$. We used this approximation, instead of the exact stochastic simulation algorithm that directly samples the dwell-time distribution[52], because the resulting trajectory is intended to serve as an input for FFT that requires equally spaced data points. Nonetheless, when a short time interval was used as in our case, the trajectories generated using the above algorithm were found to provide a relatively accurate description of the system's behavior, as we observed good agreement between their power spectra with the corresponding deterministic solutions.

**Simulation and analysis of rundown**. The trajectory of rundown was simulated using an algorithm similar to the above with the following modifications. We modeled channel rundown as a stochastic first-order decay of the number of conducting units with a decay probability in the infinitesimal time interval ($dt$),

$$p_{decay}(N_t, dt) = prob\left(N_{t+dt} = N_t - 1 | N_t\right) \overset{\Delta}{=} \lim_{dt\rightarrow0}\left(N_t k_{decay} dt\right) \tag{38}$$

An array consisting of $N$ single-channel trajectories was used to describe a channel population and the sum of these trajectories gives the ensemble current at steady-state. The number of trajectories was recorded as $N$ and the ensemble current was calculated using

$$I(t) = \sum_j i_j(t) \tag{39}$$

A random number ($r$) was drawn from the uniform distribution in the unit interval and whether a decay occurs in the next time step was assigned according to the following acceptance criterion. If $r < p_{decay}(N_t, dt)$ then $N_{t+dt} = N_t - 1$, and one of the single-channel trajectories was removed according to a random integer drawn from the discrete uniform distribution in the interval $(0, N_t)$. The ensemble current at $t + dt$ was calculated by taking the sum of the surviving single channels whether or not a decay has occurred. This procedure was then repeated until the desired length of the trajectory is reached. As an alternative mechanism, we also modeled channel rundown as a slow entry into a non-conducting state from the open state. The trajectory was simulated as in the simulation of single-channel records, but the initial occupancy of states was assigned according to those calculated for only the closed and open states to mimic rapid equilibration before the current decays. In both cases, the resulting macroscopic current is characterized

by a slow decay when $k_{decay} > 0$, which reflects the decrease in the number of activatable channels over time.

The power spectra of the simulated ensemble current with or without rundown, from the same set of single-channel trajectories, and the survival trajectory were obtained via FFT. The one-sided deterministic power spectrum of the survival trajectory, modeled as an exponential decay, was calculated using

$$G(f) = \frac{2}{T}(NP_o)^2 \frac{k_{decay}^{-2}}{1 + \left(\frac{2\pi f}{k_{decay}}\right)^2} \quad (40)$$

where $f$ is frequency, $N$ and $P_o$ are defined as above, and $T$ is the duration of the simulated trajectory.

**Poisson-Boltzmann calculations.** The electrostatic potential along a path connecting the lower $Ca^{2+}$ binding site and the cytosolic space was calculated by solving the linearized Poisson-Boltzmann equation in CHARMM[53,54] on a 240 Å × 240 Å × 260 Å grid (1 Å grid spacing) followed by focusing on a 160 Å × 160 Å × 160 Å grid (0.5 Å grid spacing). Partial protein charges were derived from the CHARMM36 all-hydrogen atom force field[55]. Hydrogen positions were generated in CHARMM. The membrane boundaries and dielectric properties of the system were as described previously[12]. The $Ca^{2+}$-free and -bound structures of mouse TMEM16A (PDB: 5OYG and 5OYB respectively) were used to represent the $Ca^{2+}$-sensing helix α6 in its resting and activated conformations respectively. Three configurations of sub-maximally bound states—resting α6 without any $Ca^{2+}$ bound, resting α6 with the upper $Ca^{2+}$ bound, and activated α6 with the upper $Ca^{2+}$ bound—were used to model the ligand-binding intermediates described in the full mechanism that was shown to adequately account for both $Ca^{2+}$ binding and gating in TMEM16A at steady-state.

**Monte Carlo estimation of confidence intervals.** $N$ single-channel records were simulated using the parameters estimated for the wild-type channel at a sampling frequency of 20 kHz. 1.5% of the records were randomly drawn from the discrete uniform distribution in the interval $(0, N)$ to mimic random sampling during an experiment. $N$ was typically 1000. The averaged power spectrum of the randomly sampled records was calculated and model parameters were estimated by minimizing the sum of squares. To account for uncertainty in the $P_O$ estimated from non-stationary noise analysis, a random number drawn from the Gaussian distribution centered at the $P_O$ of the wild-type channel with a standard deviation of 0.02 (which typically results in values ranging from ~0.77 to ~0.88 and centered at 0.82) was used as an input for each fit. This sampling procedure was then repeated 1000 times to estimate the uncertainty of each rate and equilibrium constant and their correlation.

**Statistics.** Data were collected from individual cells obtained from different transfections. Data analysis were performed using Clampfit 10.6 (Molecular Devices), Excel (Microsoft), NumPy (https://www.numpy.org), and SciPy (https://www.scipy.org). For numerical operations and simulations, NumPy and SciPy were used. Parameter optimization was performed with the least_squares function in SciPy using the described sum of squares objective function, which also computes the Jacobian matrix that was used to estimate the 95% confidence intervals. Standard error uncertainties were propagated using[56]

$$\sigma_{(a+b\,or\,a-b)} = \sqrt{\sigma_a^2 + \sigma_b^2}$$

$$\frac{\sigma_{(ab\,or\,a/b)}}{|f(a,b)|} = \sqrt{\left(\frac{\sigma_a}{|a|}\right)^2 + \left(\frac{\sigma_b}{|b|}\right)^2} \quad (41)$$

The $t$-test was used for comparison of two samples with a significance level of 0.05. Statistical analysis was performed using either Prism 8 (GraphPad) and/or NumPy/SciPy.

**Reporting summary.** Further information on research design is available in the Nature Research Reporting Summary linked to this article.

## Data availability

Data supporting the findings of this manuscript are available from the corresponding authors upon reasonable request. A reporting summary for this article is available as a Supplementary Information file. Source data are provided with this paper.

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

## Acknowledgements

We thank all members of the Dutzler lab for help at various stages of the project. This work was supported by a grant of the European Research Council (ERC no 339116, AnoBest) to Raimund Dutzler, and a Forschungskredit of the University of Zurich (grant no FK-18-048) to A.K.M.L.

## Author contributions

A.K.M.L. conceived the study, performed electrophysiology experiments and calculations, and analyzed the data. A.K.M.L. and R.D. jointly wrote the manuscript.

## Competing interests

The authors declare no competing interests.
