## [Peer Review File · Nature Communications]

Reviewer #1 (Remarks to the Author):

The work by Lam and Dutzler constitutes a very well thought-through investigation and carefully performed advanced electrophysiological analysis of the molecular mechanisms of Ca²⁺ gating in the TMEM16A channel. Some experiments are strikingly elegant in their design and execution. Furthermore, the paper is overall well written and pleasant to read.

The functional analyses described in the manuscript reveal a mechanism of channel activation where the binding of Ca²⁺ promotes conformational changes in the channel protein that in a stepwise manner couple Ca²⁺ binding to movements of the TM6 and opening of a gate in the narrow pore region. The conclusion reached by the authors constitutes an important and timely discovery. The biophysical analyses convincingly support the authors' arguments.

I invite the authors to consider carefully the feedback below (in the order it appears in the text). This constructive feedback is aimed to improve the quality of the manuscript in some instances.

- Line 60. Extended data Fig.1. The simulations of stationary current corresponding to various gating schemes show the ability of spectral analysis to pick up transitions in the gating processes. While this not an unexpected finding, I found these simulations, and Extended data Fig1 in general, useful because they enhance the clarity of the arguments posed. Because this figure aims to illustrate a concept, you may want to include a diagram of the autocovariance of the signal next to the Fourier transformation of the autocovariance plot (and explain its significance); this may help the reader to appreciate the significance of the spectra.

I take the opportunity to emphasize the strong educational value of this manuscript (in addition to the important scientific advance it provides), since it covers with significant clarity many relevant concepts in ion channel science. As you will see in my comments below, to further increase this educational value, I have suggested that some statements in the Methods are supported by additional specific references and/or slightly expanded explanation.

- Lines 80-84. A shift in the spectral frequencies as Ca²⁺ is elevated is also consistent with the hastening of the time course of current activation during stimulating voltage pulses in different [Ca²⁺]_i presented in other published papers. It may be worth mentioning this.

- Figure 1a. Please show the zero current level. I would also recommend to express the current amplitude in pA as opposed to a fraction of P_o, because it was current amplitude that was measured, the effect on P_o was based on additional measurements, and it is less direct.

- Figure 1b. It would be interesting to also see the spectra being presented non normalized, i.e. expressed in A² x s (in an additional supplementary figure), because the relative changes in power are informative. The single channel conductance could be estimated from the integral of the fitted Lorentzian(s). It would be interesting to know if the single-channel conductance changes as Ca²⁺ is raised. Indeed a point that is not fully covered in the paper is the contribution of the bound Ca²⁺ to pore electrostatic, which may modulate the "electrostatic gate" formed by the vacant Ca²⁺ binding sites. I would expect an electrostatic gate in series with a "steric" gate(s) to constitute a means to modulate the single channel conductance.

- Lines 113-115. A bit more detail on this global fitting needs to be provided. How many corner

frequencies can be reliably assessed from the fitting of spectra obtained in low Ca²⁺?

- Line 158-164. How does the spectra for I641A, I550A and I551A current in the absence of Ca²⁺ look like? Is L10 affected in 0 Ca²⁺?

- Line 165, statement "To examine whether I550A and I551A do not influence gating at all...". Given that these are constitutively open channels it is evident that the mutations have influenced gating, or they might have generated "leaky" channels. I think the statement requires some revision.

- Lines 165-184, the mutant cycle analysis works really neatly here. Lines 169-172 could be rewritten in slightly more comprehensive manner to increase clarity for readers who may not be fully familiar with the method. I found that older papers adopting this method (e.g. the classic doi: 10.1126/science.7716527) included a clearer explanation of the concepts underlying the mutant cycle analysis.

- Methods section - these are generally well written and because of this the text also has relevant educational value to e.g. final-year UG students or early PG students who wish to learn about these advanced electrophysiological approaches (and associated matrix algebra etc.). To increase this educational value even further, please do reference key steps more fully and/or slightly expand on their significance. This may include even relatively simple steps, such as the propagation of errors, a point that is occasionally overlooked by UG students - referencing a standard textbook (e.g. Taylor's Errors in physical measurements), in this exemplary case, would suffice. Please do this for all key concepts.

- Lines 337-344, molecular biology. Please state the accession number of the cDNA used. Also, please state how transfected cells were recognized (co-transfection with GFP? Etc.). Here or in the statistics section state what constituted independent electrophysiological recordings - those obtained from individual patches each obtained from a different batch of cells (i.e. separate transfection) or just individual patches regardless of the batch of cells?

- Line 353, Methods. For spectral analysis a Butterworth filter, which have a very sharp roll off, is usually preferred over a 4-pole Bessel filter. In any case, it would be useful for the reader to see (in a supplementary image) the spectrum (with ordinates in A²/Hz) of the background (i.e. obtained from the small current recorded in 0 Ca²⁺ or at the ECl) alongside the spectrum of the TMEM16A current. In this way, the reader can see whether the current noise reaches background levels at frequencies below the 10 kHz (filtering frequency). This is to ensure that no major component of high frequency channel noise will be neglected using a 10 kHz low pass filter. Another point here is that I would have preferred a slightly higher sampling rate to be used in association with a 10 kHz filtering frequency (to avoid aliasing), although your 20 kHz sampling may just be sufficient. I think the use of the Hamming window, mentioned further down in the Methods, was perfectly appropriate and I was glad to see that being mentioned.

- Lines 381-383, non-stationary noise analysis. What is the duration of the Ca²⁺ application during concentration jumps? What is the time course of changes in current amplitude during concentration jumps? To allow examination of the quality of the data, exemplary traces should be shown in the extended figure 5 alongside the computed average current and variance for a single experiment (not just the averaged variance-mean current plots).

- Line 395, unclear why variance and mean current were averaged using a Gaussian filter. Provide adequate reference for this or explain rationale.

- Line 400, I suggest to avoid relatively vague statements such as "current recorded for typically 50-100 s". The duration of the tracts of current for spectral analysis should be standardized. Furthermore, given that the current runs down, it might be preferable to consider shorter tracts of stationary currents (but of course still >> of the duration of the gating cycle), Fourier transformed with a window of defined sample points. Shorter tracts are less affected by current run down and could be averaged in succession similarly to what was done for the non-stationary noise analysis.

The description of the simulation of current rundown does not seem optimally integrated with the main text. I wonder if this part may be removed and perhaps published independently as a dedicated methodological paper?

- An "accompanying paper" was frequently mentioned in the text, but unfortunately this additional paper was not made available to me, so I cannot judge how the two papers are paired. Another piece of information that this reviewer would have liked to examine are your self-written codes (if any) that might have been used for simulations etc.

Reviewer #2 (Remarks to the Author):

The manuscript by Lam and Dutzler describes a mechanism of pore opening in the calcium-activated chloride channel TMEM16A using kinetic modeling that relies on stationary and non-stationary noise analysis. The manuscript appears to be a part of a two-manuscript series as the text mentions "accompanying manuscript".

Overall I find the use of the methodology (which itself is not novel) described in this manuscript interesting and the results that have been obtained potentially may have high importance. However, in my opinion, the approach has not been described in the sufficient detail to make judgement about its' robustness and validity.

In particular, the opening two sections of Results often gloss over some technical aspects or just refer to figure panels without going into details of explaining key concepts/results in these figures.

Major questions that arise about the kinetic model presented are the follows:

1. Why were specifically 4 states chosen in the model? I understand that one wants to simplify the problem but I was not sure why specifically 4, and why the 4 models they chose have the structural characteristics that are described? Maybe the "accompanying manuscript" makes these points clearer but from this manuscript I could not find the answers to these questions.
- 2 More importantly, how do we know that the model parameters are uniquely defined? What were the tests that have been done to show that the parameters are unique?
3. I am not familiar with Monod-Wyman-Changeux mechanism of allostery. Would be good to explain what it is.

The manuscript also describes Poisson-Boltzmann calculations of electrostatic potential in the groove area of TMEM16A. It is not very clear how the path on which the calculation was performed was generated was chosen. The snapshots given in Figure 2 (as well as in other figures) are not really informative and do not show enough structural detail necessary to understand where the pathway is.

Reviewer #3 (Remarks to the Author):

In this study, the authors address the mechanism by which the pore of the TMEM16A calcium-activated chloride channel opens in response to calcium binding. This study is an extension of an accompanying manuscript in which the authors have identified a collection of hydrophobic amino acids (Ile 550, Ile 551, and Ile 641) that govern the opening/closing process of the channel's pore. From the accompanying manuscript, the authors conclude that these amino acids function as a gate to control ion permeation through the channel. Previous cryo-EM structures of the channel in the presence and absence of calcium indicate that conformational changes occur near these amino acids (particularly associated with α -helix α 6). However, the conformations of the side chains of these amino acids in these structures are similar and the dimensions of the pore in this region (the neck) are too narrow to permit ion permeation. For this and other reasons, the authors conclude that the calcium-bound structure represents a pre-open state of the channel, in which the pore is not conductive.

This is an impressive study. The authors address the process of pore opening, that is, how the closed and pre-open states transition into a fully open state to allow ion conduction, using stationary and non-stationary noise analysis. They identify different intermediates and assign rate constants to the associated transitions. From these analyses they propose structural interpretation for the transition from the pre-open state into a fully open state. Although noise analysis is a technique that I have limited hands-on experience with, the studies appear to be extremely well executed and technically sound. The writing of the work is fairly technical in nature and the authors might consider ways to make it more accessible to readers with less expertise in this area.

From their data, the authors propose a widening of hydrophobic neck region of the pore to allow for conductive permeation through the channel. I concur with this hypothesis from a structural perspective. In their model, the transition from the pre-open state to the fully open state would involve sizable motions of alpha helices, in particular helix α 6. The authors assess the open probability of the channel to be approximately 0.85. Such a high open probability might be expected to yield a structure that represents a fully open state. The authors mention that the conditions of the structural studies may have prevented observation of this fully open state (lines 287-288), but some further discussion in this regard might be warranted. In the Discussion, the authors propose a structural interpretation of the opening of the gate (lines 302-313). From their studies, they conclude that the fully open state is more energetically stable than the pre-open state. In their model, the fully open state involves an interaction between isoleucine 551 and glutamine 649. This interaction is proposed to help stabilize the open pore. What type of interaction are the authors envisioning? I suppose the isoleucine might engage the aliphatic part of glutamine 649, but this seems to be somewhat of an unusual interaction since it would be between a hydrophobic amino acid (Ile 551) and a hydrophilic one (Gln 649). Perhaps the authors have modeled an atomic model of the fully open structure that would be informative (in addition to the model in Figure 7)?

In all, the studies markedly advance our understanding of ion permeation and gating in this unusual ion channel, the mechanisms and structure of which are distinct from other channel proteins.

We thank all reviewers for their generally positive and constructive comments, which we have incorporated in our revision and which we have addressed in detail below.

Reviewer #1 (Remarks to the Author):

The work by Lam and Dutzler constitutes a very well thought-through investigation and carefully performed advanced electrophysiological analysis of the molecular mechanisms of Ca²⁺ gating in the TMEM16A channel. Some experiments are strikingly elegant in their design and execution. Furthermore, the paper is overall well written and pleasant to read.

The functional analyses described in the manuscript reveal a mechanism of channel activation where the binding of Ca²⁺ promotes conformational changes in the channel protein that in a stepwise manner couple Ca²⁺ binding to movements of the TM6 and opening of a gate in the narrow pore region. The conclusion reached by the authors constitutes an important and timely discovery. The biophysical analyses convincingly support the authors' arguments.

I invite the authors to consider carefully the feedback below (in the order it appears in the text). This constructive feedback is aimed to improve the quality of the manuscript in some instances.

- Line 60. Extended data Fig.1. The simulations of stationary current corresponding to various gating schemes show the ability of spectral analysis to pick up transitions in the gating processes. While this not an unexpected finding, I found these simulations, and Extended data Fig1 in general, useful because they enhance the clarity of the arguments posed. Because this figure aims to illustrate a concept, you may want to include a diagram of the autocovariance of the signal next to the Fourier transformation of the autocovariance plot (and explain its significance); this may help the reader to appreciate the significance of the spectra.

We have now included the autocovariance plots in Fig. S1 and explained its significance in the section now renamed to 'Autocorrelation analysis' in the Results (line 59-66):

'The statistical properties of these fluctuations may be quantified from its power spectrum¹⁸⁻²¹, which is the Fourier transform of the autocorrelation function of the current²². For single-channel fluctuations, the autocorrelation function is characterized by an exponential decay with time constants corresponding to the system's relaxation times^{19,20}. Time intervals shorter than the relaxation times are expected to yield higher correlation, as it is more likely that the channel remains in or resamples the open state, while the correlation becomes zero at much longer times as any co-occurrence of opening events separated by these time intervals is essentially random.'

I take the opportunity to emphasize the strong educational value of this manuscript (in addition to the important scientific advance it provides), since it covers with significant clarity many

relevant concepts in ion channel science. As you will see in my comments below, to further increase this educational value, I have suggested that some statements in the Methods are supported by additional specific references and/or slightly expanded explanation.

- Lines 80-84. A shift in the spectral frequencies as Ca²⁺ is elevated is also consistent with the hastening of the time course of current activation during stimulating voltage pulses in different [Ca²⁺]_i presented in other published papers. It may be worth mentioning this.

We have added several references (8,23-25) and added the following sentence (line 95-98):

‘A shift in the spectral frequencies as Ca²⁺ concentration is elevated indicates the hastening of the activation response time in concentration^{8,23,24} and voltage jump experiments²⁵, which are governed by the same set of time constants.’

- Figure 1a. Please show the zero current level. I would also recommend to express the current amplitude in pA as opposed to a fraction of P_o, because it was current amplitude that was measured, the effect on P_o was based on additional measurements, and it is less direct.

We agree that it is often helpful to indicate the zero-current level. In this particular case, however, the current runs down over time and as a result the absolute amplitude in pA without rundown correction (that we always perform when measuring concentration-response relations) might not be meaningful and can be misleading. The way the traces are presented is based on a self-normalization to remove the contribution of the number of channels

$$I(t)/\Gamma = NiPo(t)/(Ni(Po)^{-})$$

Thus, the presentation $I(t)/\Gamma$ is equivalent to $Po(t)/(Po)^{-}$ and when multiplied by $(Po)^{-}$ gives $Po(t)$, which we think gives the best representation of the steady-state fluctuations across the concentration range.

We have added the following text to the legend of Fig. 1a:

‘The data correspond to $Po(t) = ((Po)^{-}(I(t)))/\Gamma = ((Po)^{-}NiPo(t))/(Ni(Po)^{-})$, where I is the macroscopic current, Po the open probability, N the number of channels, i the unitary current, and the bar notation indicates the mean of.’

- Figure 1b. It would be interesting to also see the spectra being presented non normalized, i.e. expressed in A² x s (in an additional supplementary figure), because the relative changes in power are informative. The single channel conductance could be estimated from the integral of the fitted Lorentzian(s). It would be interesting to know if the single-channel conductance changes as Ca²⁺ is raised. Indeed a point that is not fully covered in the paper is the contribution of the bound Ca²⁺ to pore electrostatic, which may modulate the “electrostatic gate” formed by

the vacant Ca²⁺ binding sites. I would expect an electrostatic gate in series with a “steric” gate(s) to constitute a means to modulate the single channel conductance.

We agree that the amplitudes obtained from recordings at different Ca²⁺ concentrations in non-normalized spectra can in principle provide additional information on model parameters and mechanism. However, we decided to refrain from analyzing the absolute amplitudes and used normalized spectra instead in our study because these absolute amplitudes depend on the number of channels, which in our case changes over time (due to current rundown) and is thus poorly defined. This limitation was in part overcome by using non-stationary noise analysis to estimate the single-channel current and open probability in separate experiments.

With respect to the influence of Ca²⁺ on the single channel conductance: We have previously shown that Ca²⁺ binding neutralizes an electrostatic gate imposed by the vacant binding site to enable the channel to conduct with higher capacity (reference 12 in our manuscript). Due to its high cooperativity, even at low Ca²⁺ concentration, the majority of open channels have two Ca²⁺ ions bound. For these channels, the energy barriers for anion conduction, which determine the single-channel current through the open state(s), are thus no longer influenced by increasing Ca²⁺ concentrations (which only changes the fraction of open channels and depletes the remaining open channels with sub-maximal occupancy). These observations justify the inclusion of a single conducting state in the 12-state model

- Lines 113-115. A bit more detail on this global fitting needs to be provided. How many corner frequencies can be reliably assessed from the fitting of spectra obtained in low Ca²⁺?

In our revision, we have provided more detail on the global fitting and added the following sentences to our manuscript:

line 532-534,

‘For spectra obtained at saturating concentrations, model parameters (θ) were estimated by minimizing the sum of squares between Eq. 5 (4-state Q-matrix, concentration-independent) and the experimental spectra ($y(f_j)$).’

In addition, we have analyzed the number of components using the Akaike’s information criterion (AIC) for models with different number of Lorentzians. The AIC minimum indicates the best trade-off between the goodness of fit and the number of free parameters. In the range below saturation (80 to 800nM), the highest number of components is 4-5 (obtained from the spectra at 200 nM). At saturating concentrations (2 to 4 uM), the minimum number of components required to explain the data correspond to 3. This is in general agreement with the proposed model.

We have included an additional figure to the Supplementary materials (Supplementary Fig. 5) and added the following sentence:

line 100-103:

The presence of three Lorentzian components under saturating conditions (as evaluated using by Akaike's information criterion of models with different number of components¹⁸, Supplementary Figure 5) indicates the sampling of at least four conformational states when the channel is fully occupied by Ca²⁺.

- Line 158-164. How does the spectra for I641A, I550A and I551A current in the absence of Ca²⁺ look like? Is L10 affected in 0 Ca²⁺?

In our revised manuscript we have extended the description of the kinetic states of mutants showing basal activity in their apo state. The spectra of these mutants show Lorentzian-type relaxations at zero Ca²⁺ that generally resemble those obtained at saturating Ca²⁺ concentrations. Although much higher than expected for WT, the efficacy and kinetics of the intermediate (L₂₁) and the opening (L₁₀) transitions in the apo states of these mutants are several-fold lower than in the fully Ca²⁺-bound states. We suspect that these might be in part a consequence of the inability of $\alpha 6$ to become fully activated in the absence of Ca²⁺ (which would require its partial unwinding into a π -helix conformation and its transition into a locked position) as observed in the apo structure of I551A that we present in an accompanying manuscript.

We have included a figure and a table in the Supplementary materials (Supplementary Fig. 8, Supplementary Table 3) and added the following sentences:

line 180-190:

In the absence of Ca²⁺, the corresponding power spectra of gate mutants reveal comparable transitions as observed at a saturating Ca²⁺ concentration (Supplementary Fig. 8), which indicates the sampling of states that are for energetic reasons not populated in the apo state of wild-type. The presence of three Lorentzian components suggests a similar accessibility of states under both limiting conditions (Supplementary Fig. 8d). Although much higher than for wild-type, both the efficacy and the kinetics of the pre-opening (L₂₁) and the opening steps (L₁₀) state are consistently lower in the apo than in the fully Ca²⁺-bound state, leading to a moderate reduction in the P_o^{max} (Supplementary Tables 2 and 3). This observation further confirms the role of the bound Ca²⁺ ions in influencing the energetics of the gating transitions even in mutants with considerable basal activity. This enhancement acts concomitantly with the release of an electrostatic gate that impedes anion conduction in the open state of the apo channel¹².

line 331-343:

The spectra of mutants with basal activity obtained in the absence of Ca²⁺ reveal constitutive sampling of equivalent transitions as found in the Ca²⁺-bound state thus underlining that the general mechanism of pore opening does not strictly require the presence of the ligand as expected for a mechanism that is based on a MWC model^{27,28}. Despite the similarity of the accessible states, local structural differences of the open states in presence and absence of Ca²⁺ might be expected given the strong electrostatic influence of the bound Ca²⁺ ions on the protein conformation. Such

differences have been observed in the apo structures of I551A described in an accompanying study¹³, where $\alpha 6$ adopts a partially activated conformation in which the transition from an α - to a π -helix conformation cannot be completed due to the energetic penalty of the transition and the electrostatic repulsion from the vacant binding site. Thus, although activation and pore opening can proceed without a transition into the strained π -helix conformation, the decrease in the efficacy of both the pre-opening (L_{21}) and opening steps (L_{10}) in the apo state suggests that this might have an energetic consequence on gating.

- Line 165, statement “To examine whether I550A and I551A do not influence gating at all...”. Given that these are constitutively open channels it is evident that the mutations have influenced gating, or they might have generated “leaky” channels. I think the statement requires some revision.

We have modified the sentence (line 191-192):

‘To examine how the truncation of sidechains of Ile 550 and Ile 551 influences gating, we analyzed the respective mutations in double-mutant cycles^{29,32}.’

- Lines 165-184, the mutant cycle analysis works really neatly here. Lines 169-172 could be rewritten in slightly more comprehensive manner to increase clarity for readers who may not be fully familiar with the method. I found that older papers adopting this method (e.g. the classic doi: 10.1126/science.7716527) included a clearer explanation of the concepts underlying the mutant cycle analysis

We have reworded and slightly expanded the section on double mutant cycles and referenced the mentioned manuscript.

Line 194-201:

G_{coupling} denotes the difference between the energetic effects of a mutation introduced on the wild-type protein and in the background of the other mutation. If the two perturbations are independent, its magnitude is zero as the respective backgrounds do not have an impact on the effect of the mutation. In contrast, G_{coupling} would deviate from zero if both residues interact functionally. Conversely, energetic contributions of a mutation that are independent from a particular pairwise interaction (expressed as G_{indep}) may be inferred from the effect of the same mutation introduced on the background of a second mutation where the sidechain of the interacting residue is truncated (see Methods).

- Methods section - these are generally well written and because of this the text also has relevant educational value to e.g. final-year UG students or early PG students who wish to learn about these advanced electrophysiological approaches (and associated matrix algebra etc.). To

increase this educational value even further, please do reference key steps more fully and/or slightly expand on their significance. This may include even relatively simple steps, such as the propagation of errors, a point that is occasionally overlooked by UG students - referencing a standard textbook (e.g. Taylor's Errors in physical measurements), in this exemplary case, would suffice. Please do this for all key concepts.

We have included additional references for the key steps in the methods.

- Lines 337-344, molecular biology. Please state the accession number of the cDNA used. Also, please state how transfected cells were recognized (co-transfection with GFP? Etc.). Here or in the statistics section state what constituted independent electrophysiological recordings - those obtained from individual patches each obtained from a different batch of cells (i.e. separate transfection) or just individual patches regardless of the batch of cells?

We have added the following text:

Lines 389-391:

‘Mutations were introduced with a modified QuikChange method³⁹ using the *a,c* variant of mouse TMEM16A (UniProt identifier: Q8BHY3-1) as the template and were verified by sequencing.’

Line 395:

‘Transfected cells were identified via the fluorescence of the Venus tag.’

Line 722:

‘Data were collected from individual cells obtained from different transfections.’

- Line 353, Methods. For spectral analysis a Butterworth filter, which have a very sharp roll off, is usually preferred over a 4-pole Bessel filter. In any case, it would be useful for the reader to see (in a supplementary image) the spectrum (with ordinates in A^2/Hz) of the background (i.e. obtained from the small current recorded in 0 Ca^{2+} or at the E_{Cl}) alongside the spectrum of the TMEM16A current. In this way, the reader can see whether the current noise reaches background levels at frequencies below the 10 kHz (filtering frequency). This is to ensure that no major component of high frequency channel noise will be neglected using a 10 kHz low pass filter. Another point here is that I would have preferred a slightly higher sampling rate to be used in association with a 10 kHz filtering frequency (to avoid aliasing), although your 20 kHz sampling may just be sufficient. I think the use of the Hamming window, mentioned further down in the Methods, was perfectly appropriate and I was glad to see that being mentioned.

We have included additional panels (Supplementary Fig. 7b) showing the raw power spectra extending up to 10 kHz for the recordings and their corresponding background obtained at E_{Cl} . In

all cases, the power becomes indistinguishable from the background starting from around 3-5 kHz, which justifies the use of a cutoff frequency at 10 kHz.

- Lines 381-383, non-stationary noise analysis. What is the duration of the Ca²⁺ application during concentration jumps? What is the time course of changes in current amplitude during concentration jumps? To allow examination of the quality of the data, exemplary traces should be shown in the extended figure 5 alongside the computed average current and variance for a single experiment (not just the averaged variance-mean current plots).

We have included additional panels (Supplementary Fig. 6a) showing the representative time course for both current and variance.

- Line 395, unclear why variance and mean current were averaged using a Gaussian filter. Provide adequate reference for this or explain rationale.

The use of a Gaussian filter is intended to serve as a moving average filter to average the different data points from the overlaid parabolas obtained from different patches. An example of the smoothed data overlaid with the overlaid parabolas for wild-type is shown below. Because each data pair $(\bar{I}_j, \sigma_{N_j}^2)$ is unique, a conventional averaging operation cannot be applied. However, the use of a Gaussian moving average filter on the x and y data sorted according to the x values allowed us to obtain the central tendency of the different parabolas.

We have changed lines 442-444:

Because each data pair $(\bar{I}_j, \sigma_{N_j}^2)$ is unique, they were sorted according to the \bar{I} values and were averaged using a Gaussian moving average filter, which allowed the central tendency of the normalized parabolas from different patches to be estimated.

- Line 400, I suggest to avoid relatively vague statements such as “current recorded for typically 50-100 s”. The duration of the tracts of current for spectral analysis should be standardized. Furthermore, given that the current runs down, it might be preferable to consider shorter tracts of stationary currents (but of course still >> of the duration of the gating cycle), Fourier transformed with a window of defined sample points. Shorter tracts are less affected by

current run down and could be averaged in succession similarly to what was done for the non-stationary noise analysis.

The duration of the current was standardized at 50 seconds, although in some cases a 100-second section was used to cover the region of lower frequencies.

We have changed the text (line 450-451):

‘For analysis, currents were recorded over a continuous period of typically 50 seconds, or 100 seconds for cases where sampling of lower frequencies was required.’

The description of the simulation of current rundown does not seem optimally integrated with the main text. I wonder if this part may be removed and perhaps published independently as a dedicated methodological paper?

We have provided a description of the simulations and a discussion of the effect of current rundown in the methods. It is not expanded much since, although it is an important characteristic that needed to be addressed, it is not strongly related to the mechanistic aspects that we describe in more detail in the main text.

- An “accompanying paper” was frequently mentioned in the text, but unfortunately this additional paper was not made available to me, so I cannot judge how the two papers are paired. Another piece of information that this reviewer would have liked to examine are your self-written codes (if any) that might have been used for simulations etc.

We have included several examples of the codes used in the simulations and curve fitting routines in our submission.

Reviewer #2 (Remarks to the Author):

The manuscript by Lam and Dutzler describes a mechanism of pore opening in the calcium-activated chloride channel TMEM16A using kinetic modeling that relies on stationary and non-stationary noise analysis. The manuscript appears to be a part of a two-manuscript series as the text mentions "accompanying manuscript".

Overall I find the use of the methodology (which itself is not novel) described in this manuscript interesting and the results that have been obtained potentially may have high importance. However, in my opinion, the approach has not been described in the sufficient detail to make judgement about its' robustness and validity.

In particular, the opening two sections of Results often gloss over some technical aspects or just refer to figure panels without going into details of explaining key concepts/results in these figures.

We have extended the description of the applied approach at the beginning of the results as also detailed in the response to reviewer 1. A discussion of the general procedure was included in Methods as sub-chapter ‘Validation of kinetic analysis via simulations’. The main conclusion was that the simulated single-channel trajectories and their power spectra from known mechanisms can be described by the general framework published by Colquhoun and Hawkes 1977, and that unknown parameters in the mechanism can be estimated by fitting the spectra to the mechanism, a procedure that we have validated in a simple Monte Carlo simulation.

***Major questions that arise about the kinetic model presented are the follows:
1. Why were specifically 4 states chosen in the model? I understand that one wants to simplify the problem but I was not sure why specifically 4, and why the 4 models they chose have the structural characteristics that are described? Maybe the "accompanying manuscript" makes these points clearer but from this manuscript I could not find the answers to these questions.***

As described in the response to reviewer 1, the number of relaxation components was inferred according to the Akaike’s information criterion (AIC) calculated for models with different number of Lorentzians. The AIC minimum indicates the best trade-off between the goodness of fit and the number of free parameters. At saturating concentrations (2 to 4 μM), the minimum number of components required to explain the data is 3. The presence of 3 Lorentzians is a manifestation of the presence of at least 3 non-zero eigenvalues, which is resulted from a Q-matrix of a size of at least 4x4, which therefore suggests the existence of at least 4 states.

We have added Supplementary Fig. 5 and added the following text in line 100-103:

‘The presence of three Lorentzian components under saturating conditions (as judged by Akaike’s information criterion of models with different number of components²⁶, Supplementary Fig. 5) indicates the sampling of at least four conformational states when the channel is fully occupied by Ca^{2+} .’

Additionally, we have explained the reasoning behind the structural correspondence of the states in the section ‘Mechanism of Ca^{2+} activation’ in the Results (line 106-150). Since it is quite lengthy, it is not repeated here.

2 More importantly, how do we know that the model parameters are uniquely defined? What were the tests that have been done to show that the parameters are unique?

We have calculated the 95% confidence intervals (CI) for the parameters obtained in each fit and they are documented in Table 1 and 2. In addition, we have further obtained the distributions of the parameters and the correlation amongst them using a simple Monte Carlo approach (Supplementary Fig. 3). The main conclusions from these results were that the 95% CI is often within 10% or less of the value of the best-fits and that the equilibrium constants that we analyzed do not show strong correlation with one another.

3. I am not familiar with Monod-Wyman-Changeux mechanism of allostery. Would be good to explain what it is.

MWC models are typically cyclic schemes where at a minimum a closed and an open state can exist for each level of ligand occupancy. The number of states and the level of ligand occupancy can be extended accordingly if desired. A schematic depiction of such mechanism can be found in Fig. 1d and Supplementary Fig. 2 where the number of states is 4 and the level of ligand occupancy is 3, which therefore leads to a 12-state model. We have also provided an extended description of MWC models as Supplementary Note of our accompanying manuscript.

The manuscript also describes Poisson-Boltzmann calculations of electrostatic potential in the groove area of TMEM16A. It is not very clear how the path on which the calculation was performed was generated was chosen. The snapshots given in Figure 2 (as well as in other figures) are not really informative and do not show enough structural detail necessary to understand where the pathway is.

The path was chosen to depict a plausible trajectory of Ca^{2+} -ions diffusing from the cytoplasm via the intracellular vestibule to the binding site. The start of the trajectory is located in the water-filled environment and the electrostatics here is not strongly dependent on the exact position, the end of the trajectory is defined by the position of bound Ca^{2+} in the fully Ca^{2+} -occupied state. The electrostatics at this location is important for the described argument. We have included an additional panel to illustrate the spatial relation between the path and the protein in a more distant view as Fig. 2a.

Reviewer #3 (Remarks to the Author):

In this study, the authors address the mechanism by which the pore of the TMEM16A calcium-activated chloride channel opens in response to calcium binding. This study is an extension of an accompanying manuscript in which the authors have identified a collection of hydrophobic amino acids (Ile 550, Ile 551, and Ile 641) that govern the opening/closing process of the channel's pore. From the accompanying manuscript, the authors conclude that these amino acids function as a gate to control ion permeation through the channel. Previous cryo-EM structures of the channel in the presence and absence of calcium indicate that conformational changes occur near these amino acids (particularly associated with α -helix $\alpha 6$). However, the conformations of the side chains of these amino acids in these structures are similar and the dimensions of the pore in this region (the neck) are too narrow to permit ion permeation. For this and other reasons, the authors conclude that the calcium-bound structure represents a pre-open state of the channel, in which the pore is not conductive.

This is an impressive study. The authors address the process of pore opening, that is, how the closed and pre-open states transition into a fully open state to allow ion conduction, using stationary and non-stationary noise analysis. They identify different intermediates and assign rate constants to the associated transitions. From these analyses they propose structural interpretation for the transition from the pre-open state into a fully open state. Although noise analysis is a technique that I have limited hands-on experience with, the studies appear to be extremely well executed and technically sound. The writing of the work is fairly technical in nature and the authors might consider ways to make it more accessible to readers with less expertise in this area.

We have made further explanations in the text at several places as also detailed in our response to reviewers 1 and 2 in an attempt to make the manuscript more accessible to a general audience.

From their data, the authors propose a widening of hydrophobic neck region of the pore to allow for conductive permeation through the channel. I concur with this hypothesis from a structural perspective. In their model, the transition from the pre-open state to the fully open state would involve sizable motions of alpha helices, in particular helix $\alpha 6$. The authors assess the open probability of the channel to be approximately 0.85. Such a high open probability might be expected to yield a structure that represents a fully open state. The authors mention that the conditions of the structural studies may have prevented observation of this fully open state (lines 287-288), but some further discussion in this regard might be warranted.

We agree that in light of the high probability of the channel in presence of Ca^{2+} and the observed structural changes of $\alpha 6$ the observed conformation should be close to a conducting conformation. However, it also appears plausible that for full activation, the narrow neck region might have to expand further to permit ion conduction. As detailed in our accompanying manuscript, the

expansion is presumably small as the neck remains narrow also in the open state and the transition might be impeded by the experimental conditions.

We have thus added the sentence in line 319-321:

This could be reflected in the Ca²⁺-bound structure of TMEM16A, where such a pre-open intermediate might have been stabilized in a detergent environment and the absence of bound PI(4,5)P₂, which prevents current rundown in excised patches^{36,37}.

In the Discussion, the authors propose a structural interpretation of the opening of the gate (lines 302-313). From their studies, they conclude that the fully open state is more energetically stable than the pre-open state. In their model, the fully open state involves an interaction between isoleucine 551 and glutamine 649. This interaction is proposed to help stabilize the open pore. What type of interaction are the authors envisioning? I suppose the isoleucine might engage the aliphatic part of glutamine 649, but this seems to be somewhat of an unusual interaction since it would be between a hydrophobic amino acid (Ile 551) and a hydrophilic one (Gln 649). Perhaps the authors have modeled an atomic model of the fully open structure that would be informative (in addition to the model in Figure 7)?

As an attempt to elucidate the nature of the interaction, we have mutated Gln649 to a leucine then to an alanine and analyzed them in respective mutant cycles with I551A. The coupling persists upon removing the partial charges in Q649L, suggesting that the interaction might be in part mediated by van-der-Waals interactions. Although we have not modelled an atomic model of the fully open structure, the orientation of the Gln649 sidechain in the Ca²⁺-bound structure seems to be in line with this interpretation. This conformation seems plausible given the amphiphilic nature of the Gln649 sidechain and its location at the interface of the aqueous pore and the hydrophobic protein region.

We have added the following sentence:

Line 232-235:

Examination of the stepwise mutant cycles reveals that the coupling persists when the partial charges of Gln 649 were removed on the Q649L background (Fig. 4e), suggesting that the interactions are predominantly mediated by the volume instead of the polarity of the side-chain and the open pore may thus in part be stabilized by van der Waals forces.

In all, the studies markedly advance our understanding of ion permeation and gating in this is unusual ion channel, the mechanisms and structure of which are distinct from other channel proteins.

Reviewer #1 (Remarks to the Author):

Many thanks for your careful consideration of my feedback.

I believe the revised manuscript represents a major contribution to the field.

Paolo Tammaro
(reviewer 1)

Reviewer #2 (Remarks to the Author):

All my concerns were fully addressed in the revision.

Reviewer #3 (Remarks to the Author):

All of my comments have been addressed. It is a beautiful paper.